# An Application for Through-Vial Impedance Spectroscopy (TVIS) in the Qualification of the Pirani-Gauge Assessment of the Ice Sublimation Endpoint

**DOI:** 10.3390/pharmaceutics17121542

**Published:** 2025-11-29

**Authors:** Pathum Subash Wijesekara, Kiran Malik, Paul Matejtschuk, Geoff Smith

**Affiliations:** 1DMU LyoGroup, School of Pharmacy, De Montfort University, The Gateway, Leicester LE1 9BH, UK; psw126@outlook.com; 2Analytical & Biological Sciences, Medicines & Healthcare Product Regulatory Agency, Blanche Lane, South Mimms, Potters Bar, Hertfordshire EN6 3QG, UK; kiran.malik@mhra.gov.uk (K.M.); paul.matejtschuk@mhra.gov.uk (P.M.)

**Keywords:** freeze-drying, primary drying, endpoint, amorphous, crystalline, through-vial impedance spectroscopy (TVIS)

## Abstract

**Background/Objectives:** All the industry standard methods for monitoring the freeze-drying process, from the single-vial assessment using temperature probes, such as thermocouples, to batch assessments using comparative pressure measurements, have poorly defined transitions marking the end of ice sublimation. In this study, through-vial impedance spectroscopy (TVIS) is used to characterise and validate the point on the Pirani curve that corresponds to the end of ice sublimation. The impact of the solution composition in relation to its propensity to form crystalline and amorphous domains and the impact of the batch size were investigated. **Methods:** Individual TVIS vials were placed at specific positions across the shelf, in order to represent the core and edge vials of the batch. The unique features of the high-frequency real part capacitance, with its precise sublimation endpoint-defining plateau, were then used to map the individual-vial sublimation endpoints onto the Pirani profile, with a view to predicting the batch sublimation endpoint. **Results:** TVIS vial endpoints enabled a key observation that the shape of the Pirani profile may be analysed in terms of two phases, the first being largely associated with ice sublimation and the second being associated with water desorption. Moreover, by identifying the transition point more precisely, even in the small to intermediate scale systems, we provide a scientific basis for predicting the sublimation endpoint for production-scale dryers, where Pirani sensors are already in place. **Conclusions:** Such qualification of batch sublimation endpoints would allow for earlier, confident switching to the secondary drying stage without unnecessary delay, leading to shorter cycles, reduced energy consumption, and improved utilisation of costly freeze-drying infrastructure.

## 1. Introduction

Freeze drying is still an extremely popular bioprocessing step, providing greatly enhanced stability to otherwise labile biologics [1], such as monoclonal antibodies, coagulation proteins, and high-dose therapeutics, as well as for poorly water-soluble small molecule API therapeutics [2]. Despite its utility, freeze-drying is an energy-thirsty technology (a typical freeze-drying facility consumes 2000 kWh and generates 8.5 tonnes CO_2_/day), and so when cycles run to several days, the environmental impact of using freeze-drying creates concerns about its sustainability [3]. Among the three main stages of lyophilisation, it is the primary drying stage that is the longest and therefore most energy-demanding in terms of the facility utilisation and carbon footprint. Consequently, this stage has been the focus of numerous process optimisation strategies, particularly those involving process analytical technology (PAT) applied to the determination of the end of primary drying [4].

Initially, the primary PAT available for monitoring product temperature during freeze-drying was the thermocouple [5], which has the longest history of use in this field [6]. However, thermocouples are typically inserted into a small subset of vials, within a batch that may contain hundreds or even thousands. This sparse sampling has been criticised for its limited representation, as it cannot reliably capture the heterogeneity of vial drying behaviour across the batch, where the edge vials dry quicker than the core vials [7]. Furthermore, concerns have been raised that the presence of the thermocouple itself may alter the heat transfer dynamics of the vial, potentially influencing the drying rate. Vials equipped with thermocouples tend to nucleate at higher temperatures compared with vials without sensors [8]. Higher temperatures result in fewer nuclei, which grow to a greater extent than those formed at lower temperatures, owing to the impact of higher temperatures facilitating water diffusion to the ice interface. As a result of the larger size distribution of the crystals, these vials generally dry more quickly due to reduced product resistance. These limitations raise the risk of failing to detect “slow-drying” vials, which may be at higher risk of incomplete drying or stability issues. An additional soak period of 10–30% is often included in order to account for these differences.

Pressure-rise-based technologies, such as manometric temperature measurement (MTM), have been available for several decades and involve deliberately closing the valve between the chamber and the condenser to assess the progress of water vapour sublimation [9,10,11,12]. This valve closure produces a measurable rise in chamber pressure from which sublimation rates and product temperature can be inferred. Such methods have been incorporated into some freeze-drying control systems as predictive tools for cycle optimisation. However, their implementation depends on the availability of rapidly actuated, reliable valves capable of sealing the chamber reproducibly. This presents a challenge for production-scale dryers, where such valves are often absent, slow to respond, or susceptible to mechanical wear and leakage. Additionally, frequent valve cycling can create pressure transients that disrupt the sublimation front, introduce uncertainty in readings, and complicate process control. While MTM and other pressure-rise methods can yield valuable thermodynamic insights, their application at the manufacturing scale remains limited due to these practical and mechanical constraints. Several studies describing the use of Pressure Rise and soft sensor data to monitor and control freeze-drying processes have been published [13,14,15].

By contrast, TDLAS (tuneable diode laser absorption spectroscopy) is an optical method that directly measures the water vapour concentration via wavelength-specific absorption, providing high sensitivity and temporal resolution without requiring chamber isolation or valve closures. Although TDLAS avoids many of the operational limitations of MTM, its implementation demands significant capital investment, specialised optical ports, and careful maintenance to ensure beam alignment and optical path integrity.

Thermocouples, cable-free thermosensors and spectroscopic and weighing systems are single-vial PAT technologies, whereas MTM, TDLAS and mass spectrometry gas analysis are all whole-batch PAT. Single-vial methods may miss heterogeneities between vials, especially in larger batches, whereas batch methods are averaging methods and so, again, early-drying vials or late-drying vials may not be sufficiently highlighted. Jameel et al., 2023 [13] compared existing methods for their strengths and limitations.

Production-scale freeze-dryers remain notoriously difficult environments in which to implement process analytical technologies (PATs). Physical and logistical constraints often limit the integration of sensor-based monitoring tools, particularly single-vial techniques that rely on physical wiring, such as thermocouples or impedance probes. Scaling such techniques across hundreds or thousands of vials is impractical and would result in unmanageable cabling and process interference. As a result, there is a clear need for methods that can qualify or calibrate existing, built-in technologies that are currently deployed within standard in production environments. In terms of scale-up suitability, Pirani gauges remain attractive because they are already integrated into most freeze-dryers and can provide continuous, disturbance-free monitoring—albeit with limited quantitative accuracy.

The operating principle of a Pirani gauge is that it estimates the chamber pressure based on the thermal conductivity of the surrounding gas. As the chamber pressure increases, the heated filament within the gauge loses heat more efficiently to the gas, resulting in a change in electrical resistance that is calibrated to indicate pressure. Because water vapour has significantly higher thermal conductivity than dry gases such as nitrogen, even small amounts of water vapour lead to an overestimation of the true chamber pressure. Comparative Pressure Measurement (CPM) uses this property by comparing the Pirani gauge reading with that of a capacitance manometer, which provides a pressure measurement unaffected by gas composition [16]. During primary drying, the difference between these two readings reflects the partial pressure of water vapour. As ice sublimation concludes and vapour pressure decreases, this difference narrows, and the Pirani signal reaches a plateau. This plateau is often interpreted as an indicator of the end of primary drying. However, CPM is inherently limited by its inability to differentiate between water released via ice sublimation and that released through desorption from the dried matrix, with the resulting pressure profile being shaped by a complex interplay of physical phenomena: As the ice front recedes during primary drying, it exposes a progressively larger surface area of the solid matrix from which bound water can desorb. Consequently, the Pirani pressure profile typically assumes a broadly sigmoid shape: an initial phase dominated by ice sublimation, followed by a phase increasingly influenced by water desorption. The transition between these phases—marking the completion of sublimation—is often approximated near the midpoint of the curve. However, due to the smooth and continuous nature of the Pirani signal, lacking sharp inflections or discontinuities, this transition is often ill-defined in practical settings, complicating the use of Pirani data for precise endpoint detection.

The time profile of the Pirani signal and its significance in determining the end of sublimation drying have been previously studied, comparing it with dew point [17], tuneable diode laser absorption spectroscopy (TDLAS) [18] and gas plasma spectroscopy [17,19]. Patel et al. [20] compared the shape of the Pirani end-of-cycle profile with samples removed and measured for residual moisture and showed that the mid-point of the Pirani curve corresponded to the end of ice sublimation, with the remaining part of the curve being defined by moisture desorption from the solids fraction. This distinction is particularly important when scaling up processes or adapting formulations with different water-binding characteristics. Zhou et al. [21] also compared Pirani and temperature profiles as a less time-consuming solution to modelling the primary drying process, and Fontana et al. [22] and Pisano [23] described simulation tools for predicting the primary drying endpoint in industrial development. Despite these developments, there remains an unmet need for a suitable technology that can routinely interrogate and analyse the shape of the Pirani gauge time profile with respect to the end of ice sublimation.

Through-vial impedance spectroscopy (TVIS) developed by Smith and Polygalov [24] provides a non-invasive PAT tool, which unlike thermocouples, enables real-time monitoring of a number of critical parameters, including physical changes in the shape of the ice mass [25], the progressive loss of water during sublimation while predicting the average temperature of the frozen mass, which collectively allow for the determination of the individual-vial heat transfer coefficients [26] and dry layer resistance [27]. This very broad range of outputs from a single PAT method means that TVIS is a particularly attractive PAT method.

The method involves applying a small alternating voltage across electrodes attached to the outside of the vial and recording the resulting alternating current. The impedance is then determined from the ratio of the applied voltage to the resultant current and the phase shift between the two signals. An impedance spectrum is created whereby the excitation frequency steps through a range of discrete values on a logarithmic scale. In the case of the current TVIS instrumentation, this frequency range is between 10 Hz and 1 MHz, with a number of points per decade being set to 10, in order to minimise the acquisition time for one spectrum.

Each impedance spectrum is then converted to the complex capacitance spectrum in order to observe the changes in the dielectric properties of the object, i.e., the TVIS vial and its content. During freeze-drying, it is the evolving dielectric properties of the vial contents, which are influenced by the amount of unfrozen water, the structure of the ice matrix and the progression of drying. In particular, the real part of the measured capacitance at high frequencies above that across which the dielectric relaxation of ice is observed (e.g., in our case 100 kHz) serves as a sensitive indicator of ice mass loss, largely independent of product temperature effects. The distant advantage of this method is that it allows for continuous, real-time assessment of primary drying without introducing probes into the vial headspace or product.

While TVIS requires the placement of standard vials, modified with electrodes attached to the outer-surface, the external positioning of the electrodes means that the method is minimally perturbing to the ice crystallisation and subsequent drying processes, while its flexible cables and minimal mass/volume of its electrode system facilitate its integration within both laboratory and pilot scales.

In this study, we propose a hybrid approach using TVIS to interrogate and define the point on the Pirani pressure curve that corresponds to the completion of ice sublimation, across a range of formulations and for an increasing size of batch (number of vials). By understanding the physical contributions shaping the Pirani profile, namely, the sequential dominance of ice sublimation and water desorption, we aim to qualify the time point that truly represents the transition from primary to secondary drying. If this point can be robustly identified and generalised, it could serve as a predictive marker for batch-scale drying completion in production settings. This insight opens the possibility of using the Pirani gauge not simply as a qualitative monitor, but as a semi-quantitative tool for defining critical process transitions. This, in turn, would enable earlier transitions to secondary drying and lead to shorter cycles, reduced energy consumption and increased throughput and associated capital efficiency gains.

To date, no systematic approach has been established to directly correlate non-invasive, vial-scale measurements of ice sublimation—such as those provided by through-vial impedance spectroscopy—with the characteristic but continuous Pirani pressure profiles recorded at chamber scale. This gap has limited the ability to use the Pirani gauge as a robust, semi-quantitative marker of sublimation endpoint in production environments, resulting in reliance on conservative cycle extensions or costly alternative sensors.

## 2. Materials and Methods

### 2.1. Materials and Sample Preparation

All chemicals were purchased from Merck Life Science UK Limited (Gillingham, UK) unless otherwise stated. The purity of sucrose, mannitol and L-histidine complies with PhEur. Tween 20 was sourced from BioRad (low peroxide grade product 1706531). BSA was Fraction V powder, cell culture grade from PAN-Biotech UK (Wimborne, UK). Sterile water for irrigation was from Baxter product UKF7114 (Baxter Healthcare, Zurich, Switzerland).

A mannitol-sucrose-histidine-Tween (MSHT) buffer was prepared in water for irrigation containing mannitol (4% *w*/*v*), sucrose (1% *w*/*v*), histidine (20 mM) and Tween 20 (0.02% *w*/*v*). The pH was adjusted to 6.5 using concentrated HCl or NaOH as required. Two solutions were prepared for freeze-drying: (A) 5% *w*/*v* mannitol in water for irrigation, and (B) 1% w/v bovine serum albumin in MSHT buffer. Three-millilitre aliquots were dispensed into 10 mL freeze-drying vials using a one-stepper pipette.

#### Equipment

Type I borosilicate glass vials (VC010-20C, nominal 10 mL volume; Schott, sourced via Adelphi Tubes, Haywards Heath, UK) were used for all experiments. These vials have dimensions closely matching the standard 10R format. Halobutyl Igloo stoppers for a 20 mm neck diameter (West Pharmaceutical Services) were also obtained from Adelphi Tubes. Through-vial impedance spectroscopy (TVIS) vials were manufactured by LyosenZ Ltd. (Leicester, UK). These consisted of standard glass vials with copper-foil electrodes affixed to the external surface of, and connected via, narrow-gauge coaxial cables, which are terminated by MCX connectors. Thermocouples inserted into the nearest neighbour vials (i.e., adjacent to those used for TVIS analysis) were used to predict the product temperature at the TVIS node (Figure 1). Lyophilisation was conducted using a Telstar LyoBeta 15 pilot-scale freeze dryer (Azbil Telstar, Terrassa, Spain) equipped with three shelves, each capable of holding up to 260 VC010 vials or smaller trays accommodating 113 vials per tray.

### 2.2. Freeze-Drying Procedure

A series of freeze-drying experiments was conducted to investigate the effects of vial batch size and the complexity of the formulation under laboratory-scale conditions. The complexity of the formulation, and in particular the water binding capacity of the resultant solids fraction, was investigated with a 5% *w*/*v* mannitol solution, in water, to provide the minimal binding capacity, and a 1% BSA in MSHT buffer to provide one that has greater water binding capacity through the amorphous domains of the sucrose and BSA fractions.

To assess potential scale-dependent effects, each of the two solutions was freeze-dried in three separate batch sizes comprising 113, 260 and 520 vials, respectively. The arrangement of the TVIS vials and the nearest neighbour thermocouple containing vials is shown in Figure 2. In this feasibility study, each TVIS channel was connected to a single vial positioned at a defined location (e.g., front, core, and edge) within the batch. Consequently, the variability between vials at the same location was not assessed, and no error bars are presented. While this design allows a qualitative comparison of drying behaviour across positions, a more robust statistical characterisation of measurement reproducibility will require multiple TVIS vials clustered at identical locations within the same batch to enable the calculation of standard deviations and confidence intervals for estimated endpoints. Future work will adopt this approach to quantify intra-location variability and assess the precision of the method.

In each freeze-drying experiment, the tray(s) of vials were loaded into the freeze-dryer at ambient temperature and cooled to −50 °C over a period of 90 min, followed by two successive annealing steps at −15 °C and −28 °C, respectively. Primary drying was initiated by reducing the chamber pressure to 200 µbar, while ramping the shelf temperature to 0 °C for the mannitol batches, and either 10 °C or 20 °C for the BSA/MSHT batches, at a rate of 1 °C min^−1^. After 20–25 h for the mannitol batches and up to 30 h for the BSA batches, the shelf temperature was further increased to 30 °C, over a 5 h period. Secondary drying was then conducted at 30 °C for a minimum of 10 h (an example is given in Figure 3). Process data from the freeze-drying cycle was recorded at two-minute intervals throughout the cycle.

At the end of the drying cycle, vials were back-filled to ambient pressure using nitrogen gas and stoppered in situ. Vials designated for retention were over-crimped with aluminium caps (Adelphi Tubes, Haywards Heath, UK).

Impedance spectra, with 10 points per decade over the frequency range 10 Hz to 1 MHz, and the corresponding thermocouple measurement of temperature, were acquired at an interval of 2 min, using a Sciospec impedance analyser (Sciospec GmbH, Leipzig, Germany, S/N 01-001E-0121-0A07). The analyser has 5 channels for TVIS measurement and 5 channels for thermocouple measurements. Note that the time for the acquisition of one spectrum per channel is ~10 s.

All impedance measurements were performed within the stainless-steel chamber of the freeze-dryer, which effectively acts as a Faraday cage to minimise electromagnetic interference (EMI). To further reduce electrical noise, the impedance analyser GND port was connected to the laboratory’s mains water piping (copper).

Each TVIS channel was calibrated immediately prior to use with a combination of an open circuit element and then a 100 fF reference standard (both supplied by LyosenZ Ltd. as part of their TVIS installation package). In addition, the impedance spectrum of each connected vial was verified using water for irrigation to confirm the presence of a characteristic dielectric relaxation peak before starting the experimental cycle. Any vial that failed to exhibit a characteristic impedance spectrum for pure water (i.e., showing a distinct dielectric relaxation peak in the imaginary part capacitance spectrum) was removed from the array, reconnected, and recalibrated. Only vials that demonstrated an acceptable impedance profile during this pre-run validation were included in the experiment. If calibration or connection issues persisted after recalibration, the vial was excluded from the run to avoid introducing spurious data.

### 2.3. TVIS Data Analysis

The real component of capacitance at 100 kHz was extracted from the TVIS impedance spectra (using proprietary LyoView^®^ software v1, from LyosenZ Ltd.) and plotted against time during the primary drying phase. In this case, a frequency of 100 kHz was selected, with the value of the real part capacitance thereafter defined by the term $C^{\prime }\;(100\;kHz)$. At such frequencies, the dielectric properties of ice are largely temperature independent [28,29,30] and therefore any changes to this parameter can be associated primarily with the mass of ice (rather than its temperature). That said, there is a smaller contribution from the percolation of protonic charges through the unfrozen (non-ice) fraction, otherwise known as the solids fraction.

This characteristic of the high-frequency real part capacitance is essential for reliably determining the sublimation endpoint, because as sublimation progresses, the product temperature inevitably increases. At lower frequencies, the dielectric properties of ice are more strongly dependent on temperature, which can lead to an artificial rise in the measured capacitance as the frozen matrix warms. Such an effect might obscure the transition to the plateau phase and reduce the precision of endpoint detection. By contrast, at 100 kHz, any temperature-induced changes are negligible, ensuring that the recovery of the capacitance reflects the shrinking of the ice dome across the base of the vial, rather than a temperature artifact. This frequency, or indeed any frequency between 100 kHz and 1 MHz, which is the upper limit of the frequency range of the current analyser, is therefore optimal for isolating the influence of ice mass on the signal profile.

The high-frequency, real part capacitance profile, which is a unique characteristic of this TVIS measurement, has previously been interpreted by Pandya et al. [25,30] in terms of morphological changes in the ice mass during sublimation of frozen water. This was later described in a generalised schematic [31] that is intended to capture the features of the high-frequency real part capacitance profile, during the sublimation of ice from frozen solutions contained in the TVIS vial (Figure 4). The schematic is divided into sections, with start points labelled A to E that provide a mnemonic to assist in memorising the various stages:

Point A (for “application”) marks the point when the vacuum is applied.

Point B (for ‘beginning’) is the onset of the steady state period and continues until point C. This period corresponds to the linear decrease in the ice cylinder height, which maintains intimate contact with the glass wall, and without changing the shape of the ice interface.

Point C (for “curvature”) is when the ice starts to preferentially dry down the sides of the glass wall, resulting in the perimeter of the ice interface changing from planar to a curved surface. With the continuation of drying, the interface ultimately resembles a dome. “C” also stands for the curvature of the capacitance-time profile and implies the point when the linear decrease in capacitance changes to a more curved profile.

Point D (for “dome”) is when there is no ice cylinder in contact with the glass wall and only an ice dome covers the base of the vial. “D” also stands for dip and implies the point at which the capacitance reaches its minimum value, after which it starts to increase again.

Point E (for “end”) is the end of ice sublimation, i.e., the point in time when the last trace of ice disappears.

## 3. Results

The $C^{\prime }\;(100\;kHz)$ profiles for an example core vial from each of the five lyophilisation (Figure 5 and Figure 6) demonstrate the commonalities observed for all formulations studied, in that they all display the unique profile described in Figure 4. Bear in mind that the profile described in Figure 4 was first observed for frozen water, at the front of a batch of vials, where photographic evidence captured the evolving shape of the ice mass [21]. Now, having observed a similar response from these frozen solutions, one may expect that the transition points A to E all represent the same physical changes in the ice mass that were observed for pure frozen water. The distinct advantage of TVIS, in this regard, is that it obviates the need for some form of X-ray tomography to see behind the ‘curtain’ of the dry layer, behind which the ice mass is obscured.

The physical size of the graphs in the y direction has been scaled so that the tram lines from the dip to the plateau are the same size. The additional dashed red line that tracks the trajectory of the recovery phase for the 113-vial batch has been mapped onto the 260- and 520-vial batches, in order to show the relative recovery rates between the core vials within the 113, 260 and 512 batches. In the case of 1% mannitol, these lines demonstrate that the recovery rate is greatest for the vial at the core of the 520-vial batch, with the recovery rate of the 260-vial batch being somewhere in between. However, for the 1% BSA in MSHT buffer, there appears to be almost perfect alignment of the recovery behaviour in all three core vials, from the three batches of 113, 260 and 520 vials.

With the limited dataset under scrutiny, it is not yet clear whether these differences in the rate of recovery for the mannitol batches are significant and whether they point towards some structural differences in the frozen mass.

Next, a simple method was developed for the retrospective identification of the ice sublimation endpoint, based on the assumption that the point at which the data reaches a steady-state value (or plateau) marks the effective end of ice sublimation (see point E on Figure 4). This method is illustrated and described in Figure 7 using the data for a core vial (TV3) containing 5% w/w mannitol, as an example. Note that all endpoints are defined based on the point at which the residual data diverges by more than 2.2 RMSD from zero.

For each drying run, the time points t1 and t2 defining the regression interval were selected manually based on visual inspection of the C′ (100 kHz) profiles. The aim was to identify a segment within the later part of primary drying where the signal had reached a stable plateau, as indicated by a consistently low value. While this manual selection introduces an element of subjectivity, it enabled consistent identification of the plateau region across experiments. Future work will focus on developing automated or statistical methods to define these intervals objectively and improve reproducibility.

With this general method in place, we now focus on an anomaly whereby the edge vials display an additional feature to that described in Figure 4. This feature occurs within the main recovery phase (period D to E, Figure 4) between the rather steep rise in $C^{\prime }\;(100\;kHz)$ with time, and the part of the profile in which $C^{\prime \;}(100\;kHz)$ no longer changes (i.e., the plateau region). Over this intermediate period, the rate of change in $C^{\prime }\;(100\;kHz)$ is in between that of the steep recovery phase and the plateau (see period t1 to t2 of the Pirani profiles of the edge vials in Figure 8). Interestingly, the end of this phase (i.e., the point that the data reaches a plateau at time t2) is just slightly ahead of the endpoint of the core vial, which in turn corresponds to the mid-point of the Pirani curve (also shown in Figure 8 by the vertical red lines which map the individual vial end points onto the shape of the Pirani profile) suggesting that the edge vial may be used as a predictor for the sublimation end point of the entire batch

We conducted exploratory analyses to evaluate the sensitivity of the estimated endpoint to the RMSD criterion. Specifically, increasing the threshold from 2.2 RMSD to 3.3 RMSD systematically shifted the identified endpoints earlier in time, often to positions on the plateau of the C′(100 kHz) profile that preceded the characteristic “step down” transition observed in the front vials. Conversely, reducing the threshold to 1.1 RMSD consistently delayed the estimated endpoints beyond the mid-point of the Pirani profile in core vials. This outcome conflicts with prior findings (e.g., Patel et al. [20]), indicating that the approximate mid-point of the Pirani curve represents the end of ice sublimation for smaller batch sizes. Based on these comparative observations, we selected 2.2 RMSD as a practical compromise that balances sensitivity and specificity in the context of this study.

The optimisation of the RMSD threshold used to retrospectively identify the sublimation endpoint warrants a dedicated study with a larger sample size and rigorous statistical modelling, so future work will systematically assess alternative RMSD thresholds to build a more robust statistical mapping between TVIS-determined endpoints and Pirani profiles across diverse formulations and scale conditions.

At this point in the evaluation of the data, the sublimation endpoint of the TVIS vials appears to suggest that it is the mid-point of the Pirani curve that represents the batch end of the sublimation process. Thereafter, the remaining position of the Pirani curve must be de facto a consequence of the water desorption from the solids fraction. These observations are in agreement with Patel et al. [20], who determined that it was the mid-point of a sigmoidal decrease in the Pirani profile that signified the sublimation endpoint, albeit in their case, the solution under study was 5% sucrose. It is important to note that the example shown in Figure 8 is for the smallest batch size of 113 vials, in which the Pirani profile is quite symmetrical on both sides of the mid-point. For the larger batch sizes, this symmetry is not always the case.

While the slope and symmetry of the Pirani curves were noted qualitatively, formal quantitative modelling of these features was beyond the scope of this initial study and will be addressed in future work. In addition, the quantitative assessment of the duration and slope of the desorption phase was beyond the scope of this study but will be pursued in future work to better characterise its dependence on batch size and formulation properties.

Next, we examine how the shape of the Pirani curve changes with the formulation type, from a crystalline mannitol formulation to a partially crystalline formulation containing a significant proportion of the glass forming excipients, 1% sucrose and 1% BSA, and how to map the individual TVIS vial endpoints onto the Pirani curve, in an attempt to establish a generalised relationship for the prediction of the batch sublimation endpoint. First, consider the characteristics of the Pirani profile for 1% mannitol (Figure 9).

The Pirani pressure profile reflects two mechanistic stages of mass transfer during primary drying: an initial plateau and a gradual decline associated with the end of ice sublimation, followed by a further decline due to the end of water desorption from the solids fraction.

In the case of crystalline mannitol, at the smallest batch size of 113 vials, the abruptness of the final descent is explained by the limited surface area for adsorption and the absence of amorphous domains, resulting in a minimal and short-lived desorption phase. This shape is akin to the trajectory of the plane that takes a nose-drive into the ground (the “plane crashing” analogy). The last TVIS endpoint (for channel 3 at the core of the vial array) sits close to the mid-point of the Pirani curve. However, there is no inflection or other characteristic that would otherwise point to the Pirani mid-point being that of the ice sublimation endpoint.

As the batch size increases to 260 vials and then 520 vials, the vapour pressure drop associated with the sublimation phase is quite similar across the batches. However, the characteristic vapour pressure drop associated with the desorption phase becomes more gradual (akin to a “plane landing”). This is more likely to be due to non-uniform drying dynamics (associated with the process of desorption) and less likely due to potential limitations in chamber throughput (increased vapour resistance due to crowding) or thermal homogeneity across the shelf; otherwise, the same drawn-out profile would have been observed in the first phase, leading up to the sublimation endpoint.

Interestingly, the profile for the 1% mannitol solution shows a distinct inflection at approximately 15 ¾ hours for the 520-vial batch. This point closely aligns with the end of ice sublimation observed in the TVIS core vial located on the bottom shelf (connected to channel 4 of the spectrometer). Based on the collective observations from the various TVIS channels, this inflection is interpreted as marking the completion of ice sublimation across the entire batch. Note that, in the case of 5% mannitol, the inflection occurs after the mid-point.

Next, consider the profiles for 1% BSA in MSHT buffer, for the batches of 113, 260 and 520 vials (Figure 10). In all cases, the first ‘half’ of the curve associated with the end of ice sublimation is similar across all batch sizes, and again, as with mannitol, the key differences are in the second part (the water desorption phase). Like mannitol, this second phase is characterised by a more gradual reduction in the vapour pressure, determined indirectly by the Pirani gauge. For the 133 and 260 batches, the sublimation endpoint for the core TVIS vial again seems to suggest that it is the mid-point of the Pirani profile that represents the batch sublimation endpoint. However, for the 520 batch, the profile becomes strongly asymmetric, and like mannitol, shows an inflection point close to the ice sublimation endpoint of one of the core TVIS vials, i.e., that connected to channel 2 of the analyser, which is placed on the top shelf (see downward blue arrow on Figure 10). Again, this seems to suggest that TVIS vials placed across the batch may collectively point towards the batch endpoint of ice sublimation. Note that, in the case of the BSA formulation, the inflection point occurs before the mid-point of the Pirani gauge.

Table 1 and Table 2 give the numerical values of the sublimation endpoints of the TVIS vials placed at various positions across the shelf (or shelves, in the case of the 520-batch size). Figures in bold are the core vials, with the average of these values given in row 7 of the table, and with row 8 containing the difference between this average endpoint for the core vials and the endpoint for the front vial. In the case of the 520-vial batch, there are two front vials, so the average of these was taken and that figure subtracted from the average of the sublimation endpoints for the core vials. From these results, one can see that the difference in endpoints between the core and the front vials is anything between ~2 and ~4 h, though no specific trend in relation to the formulation or batch size could be established.

In a further study, 260 vials of 5% mannitol in water (3 mL fill per vial) were lyophilised on the same cycle and the run interrupted at the halfway point in the fall in the Pirani curve, indicative of the end of primary drying (stopping when a Pirani reading of 290 µbar pressure was observed). Vials were then back-filled with inert gas, stoppered in situ and examined after being taken out of the dryer. There was no visible inhomogeneity of the cakes, nor any evidence of collapse or residual liquid apparent in the core vials identified at the centre of the tray, which might be expected if sublimation was still ongoing at this point (Figure 11). As these are the slowest vials to dry, this would be the most likely point to observe such a defect. Moisture determination on selected vials showed a relative moisture content of 2.4% *w*/*w* compared to 0.43% for vials dried through to the end of secondary drying. In addition, the general appearance of the base of the vials across the batch showed good homogeneity. This would support our hypothesis that the midpoint in the Pirani curve response is consistent with sublimation having ended.

## 4. Discussion

This study explored the feasibility of using through-vial impedance spectroscopy (TVIS) to identify the endpoint of ice sublimation during freeze-drying and to map these observations against the established Pirani pressure profiles across different formulations and batch scales. In the following sections, we discuss the interpretation of Pirani curve features in relation to sublimation and desorption, the influence of formulation composition on profile characteristics, the mechanistic basis for using the Pirani mid-point as an indicator, the limitations associated with scale-up and sampling representativeness, the challenges of cross-validation with other process analytical technologies (PAT) and the broader context of emerging non-invasive PAT tools. Finally, we outline the feasibility of this work and highlight areas for future research.

### 4.1. Pirani Profiles and Batch Size

There are dramatic differences between all Pirani profiles, with certain trends developing as the batch size increases and the formulation shifts from the entirely crystalline form of mannitol (which has been annealed to ensure full crystallinity) to a form containing partially amorphous domains. The point on the Pirani curve that marks this event is shown by the vertical dashed line and labelled on all charts in Figure 9 and Figure 10 as the “Transition Point.” Relatively simple profiles resembling an asymmetric sigmoid curve are observed when the batch size is small in relation to the capacity of the dryer (i.e., 113 vials in a dryer comprising three shelves, with a capacity of 780). This asymmetry in the small batch size Pirani profiles is more pronounced for mannitol than the BSA/MSHT formulation. The suggestion is that the crystalline mannitol presents a much-reduced surface for water adsorption relative to an amorphous material (for example, sucrose and BSA), and therefore, one expects a dramatic reduction in the Pirani signal following the end of ice sublimation compared with the same batch size of BSA in MHST. In either case, there is a smooth transition between the two phases of the profile, with nothing to indicate the point at which the batch sublimation endpoint finishes. Time points from TVIS-determined sublimation endpoints suggest that it is the mid-point of the Pirani curve that corresponds to the batch endpoint of sublimation. This is in agreement with Patel et al. [8] in their study on 5% sucrose. However, as the batch size increases, an inflection in the profile (coupled to the observed TVIS-determined sublimation endpoints) suggests that the profile may then be divided into two distinct regions, with the first being dominated by sublimation and the second by desorption. The second half of the profile is unusually extended as the batch size increases, and interestingly, this occurs for both the mannitol formulation (presumed to be entirely crystalline) and the partially amorphous BSA/MSHT material. Then, we reiterate that the sublimation endpoints (inflections in the Pirani profile) for the mannitol and BSA formulations occur after and before the mid-point, respectively.

### 4.2. Influence of Formulation Properties

While this study included both fully crystalline formulations (mannitol) and more complex, partially amorphous systems containing sucrose, histidine, Tween 20 and protein, it is important to note that the clarity of the transition between ice sublimation and subsequent water desorption phases in the Pirani profile may be influenced by formulation-specific properties, including the extent of amorphous content, water-binding interactions and residual moisture distribution. In particular, highly heterogeneous or glass-forming formulations may exhibit more gradual or less distinct transitions, complicating endpoint interpretation. Our observations suggest that larger batch sizes enhance the visibility of this transition point. However, additional studies across a broader range of excipients and process conditions will be necessary to establish the generalizability and robustness of this approach for endpoint detection in industrial settings.

### 4.3. Mechanistic Basis of Pirani Mid-Point Interpretation

It is important to note that the suggestion of interpreting the approximate mid-point of the Pirani profile as corresponding to the end of ice sublimation is primarily based on earlier empirical studies, particularly the work of Patel et al. [20], who demonstrated this relationship in sucrose formulations using direct sampling and moisture assays. Mechanistically, this convention is rationalised by the principle that ice sublimation must necessarily precede the onset of significant water desorption, as desorption requires exposure of the dried matrix surface. The present results demonstrate that the Pirani gauge profile is a complex convolution of sublimation dynamics, desorption kinetics and evolving product resistance. A definitive mechanistic model capable of predicting the precise transition point would require simulation approaches, coupled with detailed data on product microstructure and porosity, and is beyond the scope of this study.

### 4.4. Limitations of Scale and Industrial Implementation

Although this study was conducted in a pilot-scale freeze-dryer with up to 520 vials (representing two full trays), we recognise that this scale does not fully capture the complexity, heterogeneity and operational constraints encountered in large production dryers. In particular, the integration of wired TVIS instrumentation into a GMP-scale environment presents substantial challenges due to cable routing, shelf-loading mechanisms and compliance requirements. While our findings provide novel insights into the evolution of Pirani profiles and the correlation of TVIS-determined endpoints across increasing batch sizes, further validation studies on larger equipment will be essential to confirm the applicability of these observations under full manufacturing conditions. We anticipate that the conceptual framework and methodologies developed here can serve as a foundation for future work in production settings, contingent on appropriate adaptations to accommodate large-scale operational requirements.

Studies at the manufacturing scale may only be possible within those limited time windows that may exist, between the manufacture of a new industrial-scale dryer and its installation and qualification, which would, if at all possible, require the cooperation of both the manufacturer of the dryer and the organisation purchasing the equipment.

### 4.5. Sampling Limitations

While TVIS provides a less invasive approach than thermocouples for monitoring primary drying dynamics, it still requires careful handling and positioning to achieve consistent electrical contact and reproducible measurements. Moreover, like all single-vial monitoring techniques, TVIS is inherently limited by the number and placement of instrumented vials, which may not fully capture heterogeneity across large batches or production-scale equipment. In this study, we selected vials positioned in the front and core regions of the shelf to illustrate known differences in drying behaviour resulting from radiative heat transfer and edge effects. Future work will include clustering multiple TVIS channels in representative core positions to improve statistical confidence and may expand sampling further by deploying multiple analysers, recognising that such configurations introduce additional logistical complexity related to cabling and system integration. A Figshare repository has been set up to capture results from a broad range of studies, which demonstrate, in general, that under “normal” conditions resulting in an adequately porous dry layer (i.e., one that has not collapsed), the TVIS endpoints all sit close to the mid-point of the Pirani curve.

### 4.6. Cross-Validation with Established PAT Methods

Although several established techniques for monitoring freeze-drying progress—such as manometric temperature measurement (MTM), pressure rise testing and dewpoint humidity analysis—can provide complementary information about the evolution of water vapour in the chamber, all of these methods fundamentally detect total water vapour flux and therefore cannot differentiate between ice sublimation and subsequent moisture desorption from the dried matrix. This limitation reduces their utility for cross-validating the sublimation-specific endpoint identified by TVIS. While preliminary investigations with dewpoint humidity monitoring are underway in our laboratory, the work has not yet progressed to a stage that would allow robust comparison or inclusion here. A more definitive approach would be to interrupt the cycle shortly after the TVIS-determined endpoint and examine the product visually for melt-back or residual ice, but this strategy requires mechanically robust crystalline formulations and dedicated experimental design. We plan to pursue such validation studies in future work.

### 4.7. Comparison with Emerging Non-Invasive PAT Tools

In addition to the techniques evaluated here, a range of emerging non-invasive PAT tools have been proposed for freeze-drying monitoring. Wireless temperature sensors, such as the Tempris^®^ system (Holzkirchen, Germany) offer the advantage of real-time temperature tracking within the product without the cabling challenges of traditional thermocouples, and have shown promise in both pilot and manufacturing scales. However, the temperature profile observed as the vial warms up (much like the Pirani profile) presents as a broad curve, which asymptotically reaches a plateau when the contents of the vial reach thermal equilibrium with the shelf. This broad nature of the re-heating warming profile is also an inevitable consequence of the intrinsic overlap between the end of ice sublimation and the developing process of moisture desorption. The point at which the profile across the shelf temperature is often considered the end of ice sublimation, but this is a purely arbitrary standpoint.

Advanced spectroscopic methods, including near-infrared spectroscopy and Raman techniques, have also been explored for monitoring drying dynamics, but their integration into production-scale equipment presents substantial engineering and validation challenges. The actual size dimensions of the sheathed fibre optic probes mean that there will be inevitable consequences for the packing of the vials if these sensors are brought into close proximity to the target vials.

By contrast, TVIS provides a unique combination of non-invasive measurement, sensitivity to the evolving ice mass and scalability across batches without requiring specialised optical access or proprietary vial designs. The approach, therefore, occupies a complementary niche within the broader PAT landscape, bridging the gap between temperature-based sensors and indirect chamber pressure measurements.

### 4.8. Feasibility and Future Research Directions

The present study was designed as a feasibility investigation to explore qualitative correspondences between TVIS-derived endpoints and characteristic features of the Pirani pressure profile. As such, no formal statistical treatment of replicate measurements or detailed error analysis was performed. Future work will incorporate clustered replicates, endpoint reproducibility studies and quantitative modelling to strengthen the statistical confidence in the proposed interpretations.

## 5. Conclusions

Results to date show promise that TVIS may qualify the point on the Pirani curve that signifies the end of ice sublimation across the batch, and therefore the point at which desorption of water from the solids fraction alone is responsible for the water vapour concentration in the drying chamber. This study provides the first systematic demonstration, across multiple formulations and batch sizes, that the endpoint of ice sublimation determined by TVIS corresponds either to the approximate mid-point or, in larger batches, to an identifiable inflection on the Pirani pressure profile. Importantly, the observations indicate that this inflection does not necessarily align with the mid-point. For the larger batches, the second part of the Pirani profile is characterised by a long, drawn-out tail resulting from moisture desorption. If the inflection occurs before the mid-point (as in the BSA formulation studied here), reliance on the mid-point necessarily delays the decision point for switching to the secondary drying ramp and results in longer-than-necessary cycles. Notably, the extended second part of the Pirani curve is evident even in materials, such as mannitol, that crystallise and thus typically carry less adsorbed water, although in this case, partial hydrate formation may contribute to the effect. This behaviour is not necessarily apparent for smaller batches of mannitol, which show a precipitous decrease after the mid-point. Our study, therefore, suggests a basis for interpreting Pirani signals as semi-quantitative markers of sublimation completion without recourse to invasive probes, thereby addressing a critical gap in freeze-drying process monitoring at scale. Extending these observations to larger commercial dryers could allow optimisation of production processes and reduce reliance on conservative waiting periods to de-risk the switch to secondary drying.

## Figures and Tables

**Figure 1 pharmaceutics-17-01542-f001:**
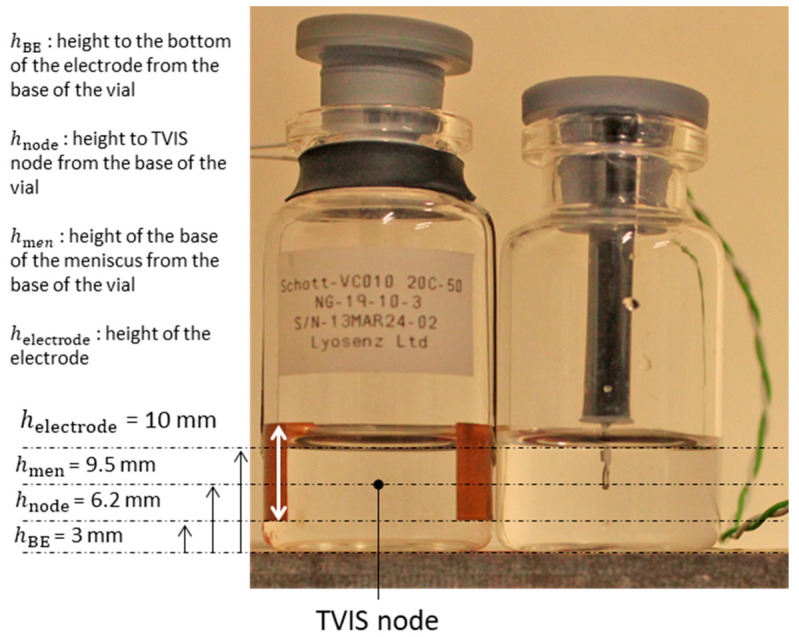
LEFT: a through-vial impedance spectroscopy (TVIS) vial, manufactured by LyosenZ Ltd., comprising a Type I borosilicate glass vial (VC010-20C) modified with a pair of copper foil electrodes (19 × 10 mm) positioned with the lower edge of the electrode, 3 mm from the base of the vial. RIGHT: thermocouple containing vial with the bead placed at a height corresponding to the height of the TVIS node.

**Figure 2 pharmaceutics-17-01542-f002:**
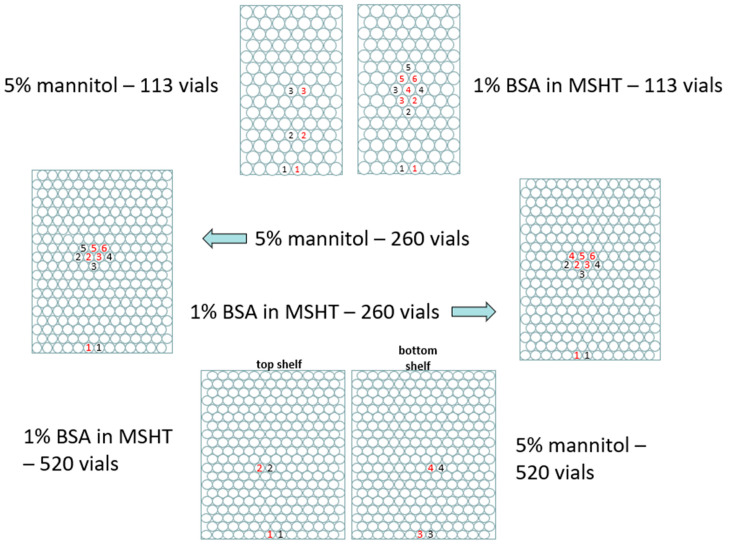
Arrangement of vials for the various batch size configurations, with the positions of the TVIS vials and the nearest neighbour thermocouple containing vials shown by numbers, 1–6, where each number represents the specific channel on the TVIS spectrometer. The colour black signifies the TVIS channel, and the colour red signifies the thermocouple channel on the TVIS spectrometer. There were 5 channels on TVIS in each run, whereby 1–6 refer to the particular vials used. The two 520 vial runs used the same layout.

**Figure 3 pharmaceutics-17-01542-f003:**
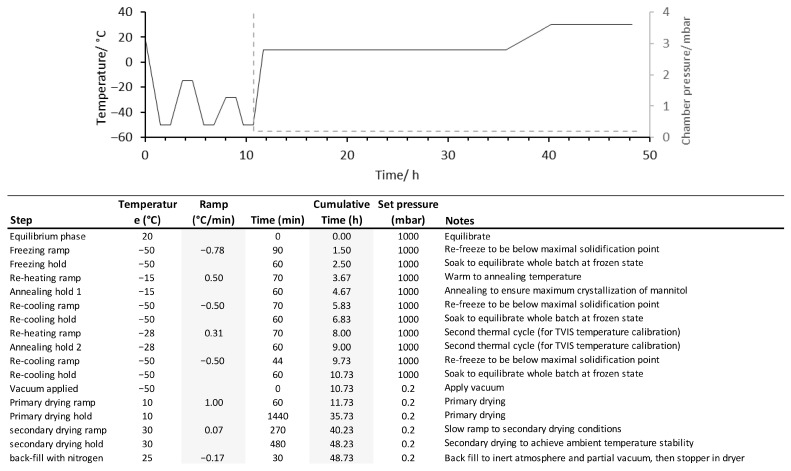
Example of a lyocycle used for freeze-drying of 1% BSA in MSHT buffer.

**Figure 4 pharmaceutics-17-01542-f004:**
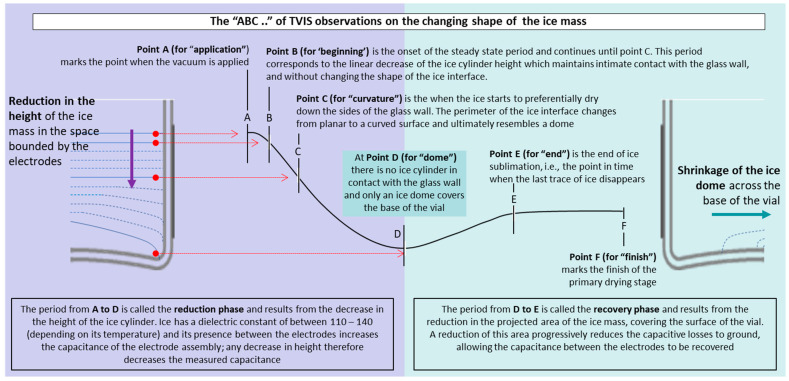
The ABC of the changing shape of the ice mass as determined by the time profile of the real part capacitance at the high frequency of 100 kHz.

**Figure 5 pharmaceutics-17-01542-f005:**
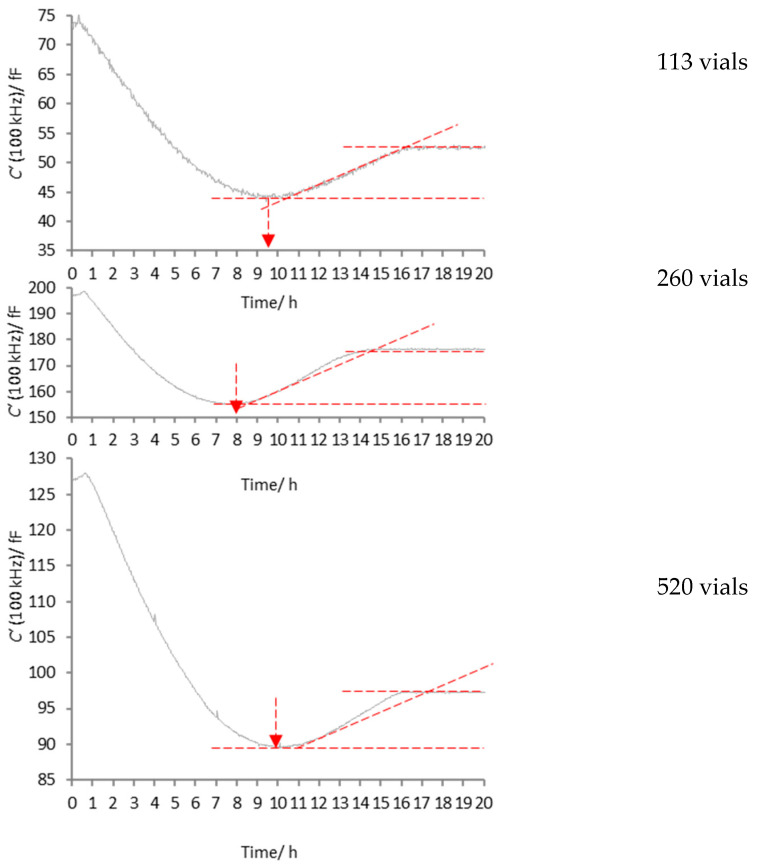
Time profiles of $C^{\prime }\;(100\;kHz)$ during the primary drying stage of the core vial containing 5% *w*/*v* mannitol solution, for the batch sizes of 113 (**top**), 260 (**middle**) and 520 vials (**bottom**). The vertical arrow shows the time point when the profile reaches the minimum capacitance which corresponds to the point in time when the ice mass is considered to take the form of a dome that covers the base of he vial, and with no contact with the side walls. The sloped dashed red line marks the trajectory of the recovery phase for the 113-vial batch, and mapped onto the profiles for the 260 and 520 vial batches in order to demonstrate that the rate of recovery increases with the batch size.

**Figure 6 pharmaceutics-17-01542-f006:**
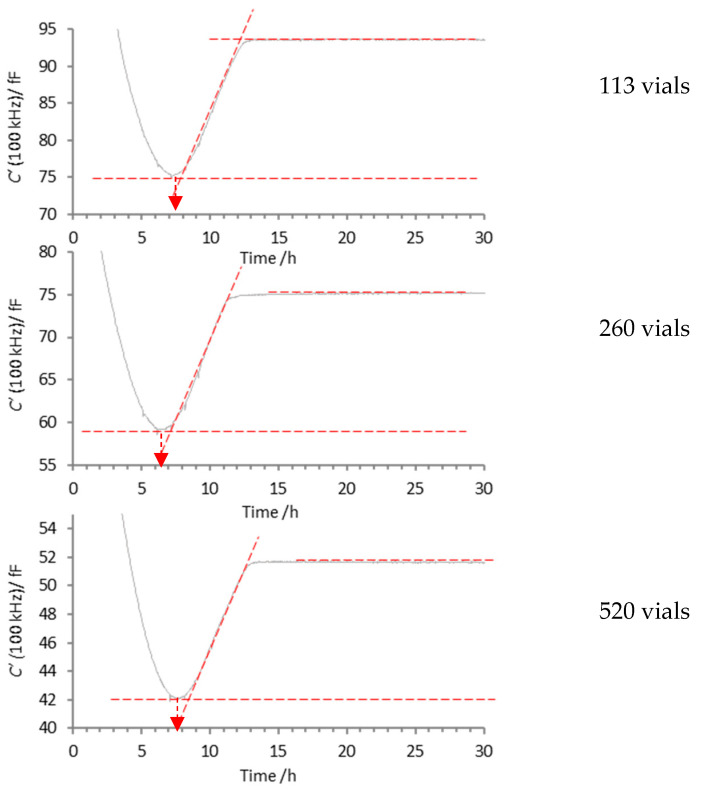
Time profiles of $C^{\prime }\;(100\;kHz)$ during the primary drying stage of the core vial containing 1% *w*/*v* BSA in MSHT buffer, for the batch sizes of 113 (**top**), 260 (**middle**) and 520 vials (**bottom**). The vertical arrow shows the time point when the profile reaches the minimum capacitance which corresponds to the point in time when the ice mass is considered to take the form of a dome that covers the base of he vial, and with no contact with the side walls. The sloped dashed red line marks the trajectory of the recovery phase for the 113-vial batch, and mapped onto the profiles for the 260 and 520 vial batches in order to demonstrate that the rate of recovery is the same for all batch sizes.

**Figure 7 pharmaceutics-17-01542-f007:**
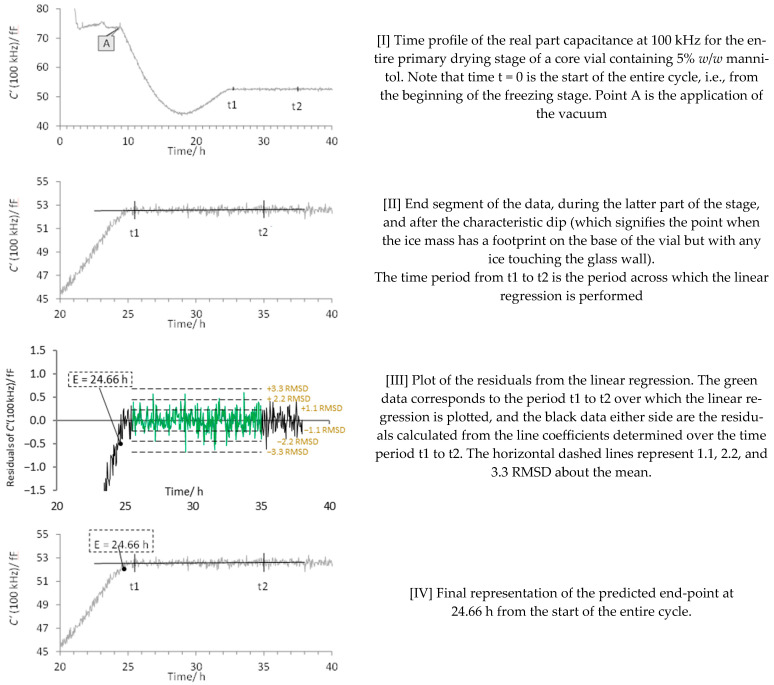
Methodology for the determination of the ice sublimation endpoint, whereby a linear regression to a selected segment of data on the plateau (between t1 and t2) allows for an estimation of the RMSD in that portion of the data, followed by tracking backwards in time to the point when the residuals in the earlier portion of data (before t1) first falls below 2.2 RMSD.

**Figure 8 pharmaceutics-17-01542-f008:**
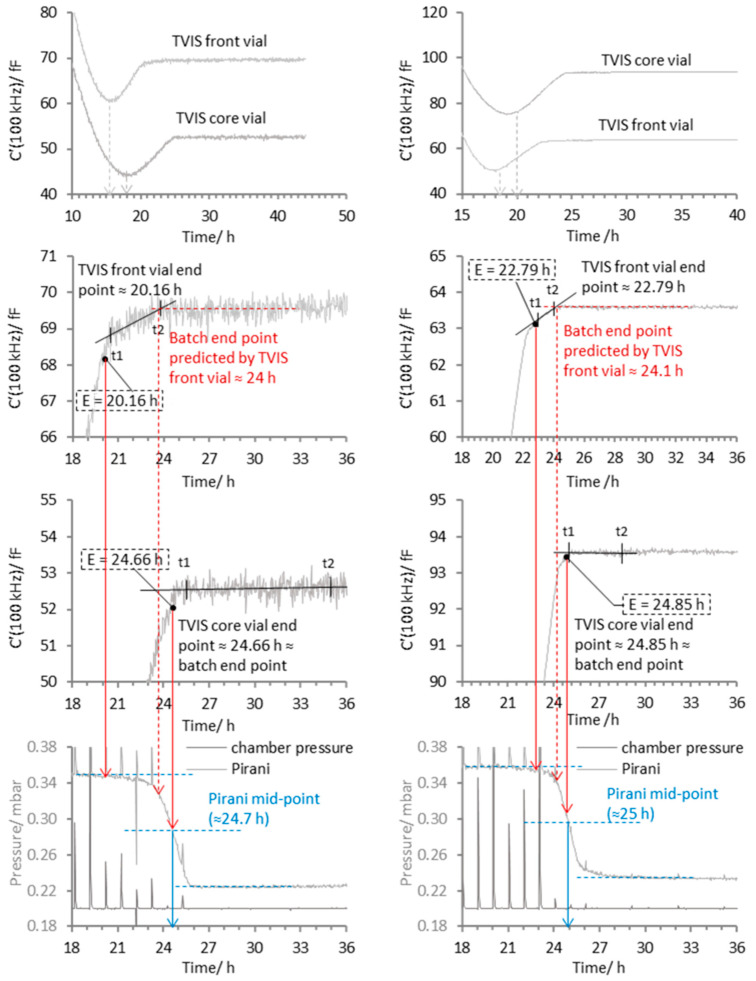
Example comparison between the core and edge TVIS vial for 5% *w*/*v* mannitol and 1% BSA in MSHT buffer for the 113-vial batch. In each case, the sublimation endpoint for the edge vial is witnessed by an inflection to a shallower gradient towards the end of the recovery phase. The vertical blue lines show the time position of the mid-point in the descent in the Pirani gauge profile. The red lines map the time points from the individual TVIS sublimation end points onto the batch profile from the Pirani gauge.

**Figure 9 pharmaceutics-17-01542-f009:**
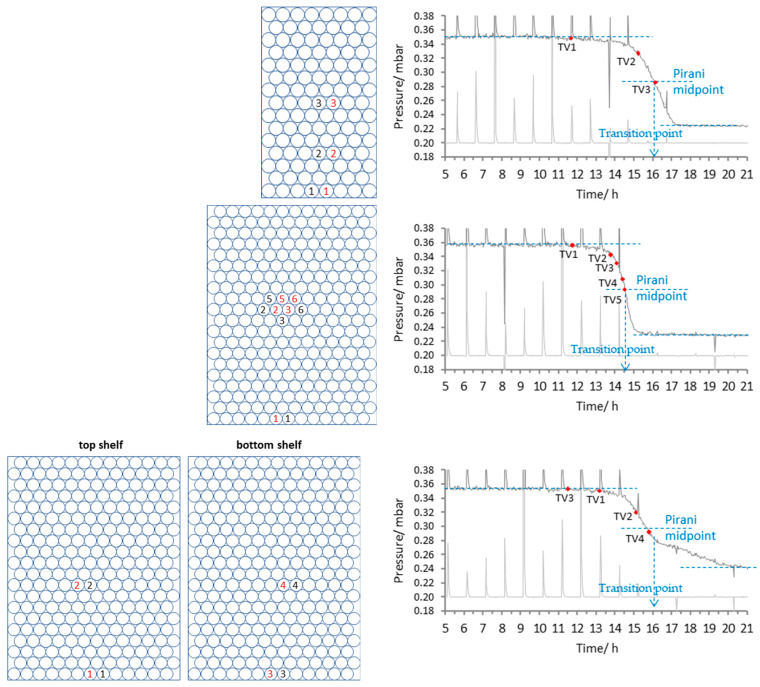
Pirani profiles for 1% mannitol (in water for irrigation) for the batch sizes of 113, 260 and 520 vials. Note that the time is normalised to the vacuum application time point and that the RMSE for the determination of the ice sublimation endpoint by TVIS was set to 2.2. Numbers in red are the TC containing vials and numbers in black are the TVIS vials. The vertical dashed blue lines show the time position of the mid-point in the descent in the Pirani gauge profile. The red dots map the time points from the individual TVIS sublimation end points onto the batch profile from the Pirani gauge.

**Figure 10 pharmaceutics-17-01542-f010:**
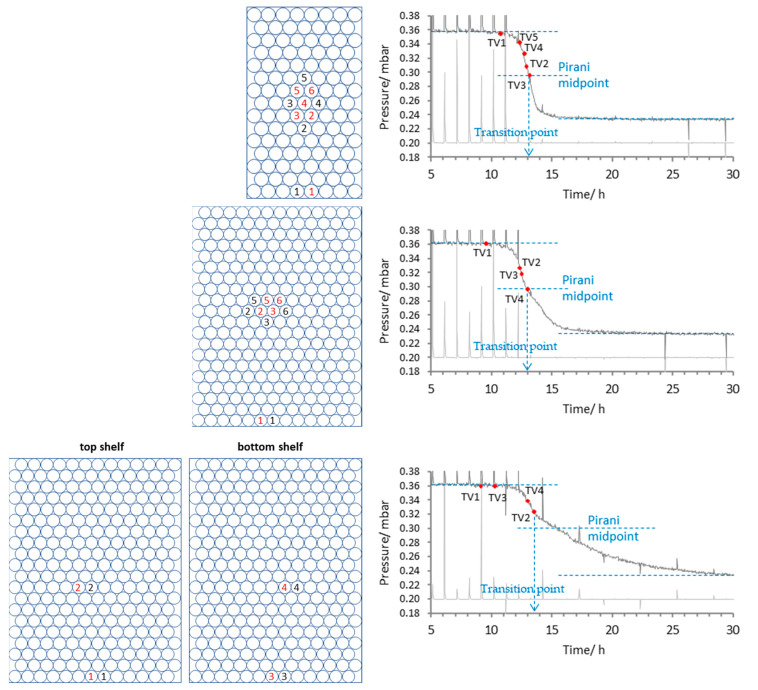
Pirani profiles for 1% BSA (in MSHT) for the batch sizes of 113, 260 and 520 vials. Note that the time is normalised to the vacuum application time point and that the RMSE for the determination of the ice sublimation endpoint by TVIS was set to 2.2. Numbers in red are the TC containing vials and numbers in black are the TVIS vials.

**Figure 11 pharmaceutics-17-01542-f011:**
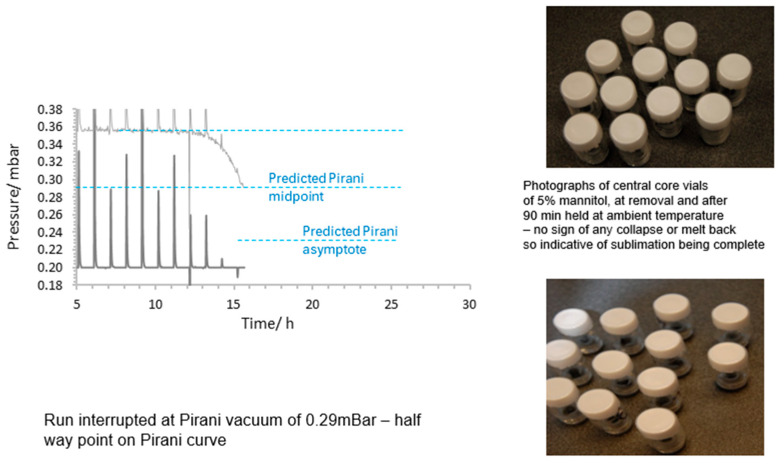
Interruption study with 260 vials of 5% mannitol in water (3 mL fill per vial) on Lyobeta 15 freeze dryer. The run was interrupted at mid-point of the previously determined inflexion curve in the Pirani profile (0.29 mBar). Vials were back-filled with nitrogen and stoppered. The base of the central 12 vials was then photographed and showed no heterogeneity or collapse area at time 0 (**upper photo**) or after 90 min (**lower photo**).

**Table 1 pharmaceutics-17-01542-t001:** Sublimation endpoints in hours for the TVIS vials containing 5% mannitol in batch sizes of 113, 260 and 520 vials.

TVIS Vial	113 Vials (EP in h)	260 Vials (EP in h)	520 Vials (EP in h)
1	11.63	11.69	13.16
2	15.26	13.79	15.09
3	16.13	14.09	11.49
4		14.43	15.93
5		14.49	
Av. of core vials	15.70	14.2	15.51
Difference	4.07	2.51	3.19

**Table 2 pharmaceutics-17-01542-t002:** Sublimation endpoints in hours for the TVIS vials containing BSA formulation in batch sizes of 113, 260 and 520 vials.

TVIS vial	113 vials (EP in h)	260 vials (EP in h)	520 vials (EP in h)
1	10.89	9.65	9.19
2	12.95	12.45	13.49
3	13.05	12.69	10.25
4	12.85	13.02	13.02
5	12.15		
Av. of core vials	12.75	12.72	13.26
Difference	1.86	3.07	3.54

## Data Availability

The original contributions presented in this study are included in the article/Appendix A. Further inquiries can be directed to the corresponding author.

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
