# Peer review of "An Application for Through-Vial Impedance Spectroscopy (TVIS) in the Qualification of the Pirani-Gauge Assessment of the Ice Sublimation Endpoint"

_pharmaceutics, 2025, doi:10.3390/pharmaceutics17121542_

Round 1

Reviewer 1 Report

Comments and Suggestions for Authors

- Weaknesses may include limited scale since the research is performed under laboratory-scale conditions (up to 520 vials), which may not entirely reflect the complexity and variability of industrial-scale freeze-drying. The authors recognize the difficulty of scaling single-vial techniques; however, additional discussion or preliminary data on larger systems would enhance the manuscript.

- Although TVIS system is less invasive than thermocouples, it still requires special handling and may not accurately represent all vials in a batch, particularly in large-scale or highly heterogeneous systems.

- The manuscript suggests that the Pirani gauge profile can be divided into ice sublimation and water desorption phases, with a distinct transition point. Nevertheless, the actual separation of these phases may be less clear in formulations with complex water-binding characteristics or in highly variable batch environments. More evidence is needed to support the robustness of this interpretation across various products and process conditions.

- While the manuscript justifies its emphasis on TVIS and Pirani gauges, it could be improved by including a more critical comparison with emerging non-invasive PATs (e.g., wireless temperature sensors, advanced spectroscopic methods

- Specific statistical treatment of results, error analysis, or reproducibility across replicates would strengthen the manuscript.

Author Response

Reviewers comment 1.1 Weaknesses may include limited scale since the research is performed under laboratory-scale conditions (up to 520 vials), which may not entirely reflect the complexity and variability of industrial-scale freeze-drying. The authors recognize the difficulty of scaling single-vial techniques; however, additional discussion or preliminary data on larger systems would enhance the manuscript.

Our response 1.1 : The study has been conducted in a large laboratory (pilot) dryer (0.5m2 shelf)  but given the size of the vials (10 ml volume) the upper scale tested (520 vials) represented two full trays. Given that the TVIS junction box (in its current design) has to be placed on the middle shelf, it follows that vials could not easily be loaded on this shelf.

We agree some validation at a small production/larger pilot dryer would have been ideal. This in more general terms is a factor that impacts a lot of the published literature on PAT developments.

However, we don’t have a larger dryer available to us within a R&D environment and our production scale dryers aren’t accessible to run such trials (that is also a common issue when trying to deploy PAT for experimental research in production scale dryers in industry).

We would stress that this is the first time such types of studies have been undertaken and that the data we obtained to date - even on these relatively small scales - shows some interesting observations with respect to the shape of the Pirani profiles that evolve as the batch size increases, which  have not been previously noted. The broad ‘concepts’ that we have developed are eminently transferrable to industrial scale studies and we believe that it will be up to industry to take on this concept and explore the opportunities for shortening the cycle duration that these methods enable.

However, we also admit that TVIS is difficult to install on a very large dryer given the cable lengths and integration logistics. At GMP scale one cannot do this – most have autoloading systems which are incompatible with any prewired probe version of PAT. Even the Tempris system require robotic systems. We makes these points on lines 105 to 111 of the original manuscript.

If it were possible to use a larger dryer we would be interested in seeing how the trends observed transferred to this scale, as we say in lines 402-4 of the original manuscript, but for now are trying to prove the principles at a moderate scale.

We have added the following to the Discussion 572-588

Limitations of Scale and Industrial Implementation Although this study was conducted in a pilot-scale freeze-dryer with up to 520 vials (representing two full trays), we recognize that this scale does not fully capture the complexity, heterogeneity, and operational constraints encountered in large production dryers. In particular, the integration of wired TVIS instrumentation into a GMP-scale environment presents substantial challenges due to cable routing, shelf loading mechanisms, and compliance requirements. While our findings provide novel insights into the evolution of Pirani profiles and the correlation of TVIS-determined endpoints across increasing batch sizes, further validation studies on larger equipment will be essential to confirm the applicability of these observations under full manufacturing conditions. We anticipate that the conceptual framework and methodologies developed here can serve as a foundation for future work in production settings, contingent on appropriate adaptations to accommodate large-scale operational requirements.

Studies at the manufacturing scale may only be possible within those limited time windows that may exist, between the manufacture of a new industrial scale dryers and its installation and qualification; which would, if at all possible, require the cooperation of both the manufacturer of the dryer and the organization purchasing the equipment.

Reviewers comment 1.2  Although TVIS system is less invasive than thermocouples, it still requires special handling and may not accurately represent all vials in a batch, particularly in large-scale or highly heterogeneous systems.

Our response 1.2 : We appreciate that the TVIS technology does require some experience in order to fit it  reproducibly however we would point out that the disruption in the vial pack is much less than for other individual vial PAT methods.

Heterogeneity across a freeze dryer shelf is especially pronounced in terms of the edge vs central vials, specially where in this case the freeze dryer door is Perspex and so much more prone to radiative heat transfer effects This is why we focussed on front row vs  middle row  vials in our studies. All single vial technologies have the draw-back of limited sampling, especially when it comes to larger scale dryers and larger batch sizes. Having demonstrated the concepts for monitoring the generalised differences between front and core the next logical steps would be to focus all the 5 channels on a selection of core vials with the target being to more precisely qualify the end point on the Pirani curves. It’s also possible to have multiple analysers, to increase the sampling to 10, 15 etc. though cable management becomes more of an issue.

We had added the following to the discussion (lines 590-601)

Sampling limitations While TVIS provides a less invasive approach than thermocouples for monitoring primary drying dynamics, it still requires careful handling and positioning to achieve consistent electrical contact and reproducible measurements. Moreover, like all single-vial monitoring techniques, TVIS is inherently limited by the number and placement of instrumented vials, which may not fully capture heterogeneity across large batches or production-scale equipment. In this study, we selected vials positioned in the front and core regions of the shelf to illustrate known differences in drying behaviour resulting from radiative heat transfer and edge effects. Future work will include clustering multiple TVIS channels in representative core positions to improve statistical confidence and may expand sampling further by deploying multiple analysers, recognizing that such configurations introduce additional logistical complexity related to cabling and system integration.

Reviewers comment 1.3 The manuscript suggests that the Pirani gauge profile can be divided into ice sublimation and water desorption phases, with a distinct transition point. Nevertheless, the actual separation of these phases may be less clear in formulations with complex water-binding characteristics or in highly variable batch environments. More evidence is needed to support the robustness of this interpretation across various products and process conditions.

Our response 1.3 : We tried to address any formulation effects by  running both single formulation runs  (sucrose or mannitol) and also a highly complex formulation with four components ( mannitol, sucrose, histidine Tween 20) together with a protein “active” as a fifth component.  Given the variability  in spontaneous  crystallisation  in mannitol-containing formulations we used  an annealing step to ensure full crystallization.

So while we accept that formulation may influence the effects we are  observing we do believe we have addressed a reasonably practical range of  formulants.

The key observation we have made, which makes the work on special interest, is that the distinct transition point between the end of ice sublimation and the continuation of the desorption process only becomes observable at the larger batch sizes. Hence we expect this to propagate into even larger batch sizes, which to some extend then addresses the reviewers first point. It would be a purely speculative suggestion that this may occur but one that doesn’t feel unreasonable.

For the smaller batch sizes we have demonstrate that mannitol has a very limited second part to the Pirani curve, one that resembles a plane crashing, whereas amorphous formulations have a more extended second part, more like a plane landing. However, when the batches size of mannitol is increased we observed, quite unexpectantly, that the second part extended quite significantly and in some sense began to resemble that of the more complex formulation

Further studies may be possible with other formulations but was not practical in the present timescale available to us on this grant-funded research.

We have included the following in the new discussion section (lines 546-557)

Influence of Formulation Properties While this study included both fully crystalline formulations (mannitol) and more complex, partially amorphous systems containing sucrose, histidine, Tween 20, and protein, it is important to note that the clarity of the transition between ice sublimation and subsequent water desorption phases in the Pirani profile may be influenced by formulation-specific properties, including the extent of amorphous content, water-binding interactions, and residual moisture distribution. In particular, highly heterogeneous or glass-forming formulations may exhibit more gradual or less distinct transitions, complicating endpoint interpretation. Our observations suggest that larger batch sizes enhance the visibility of this transition point, potentially due to greater cumulative vapor load and improved signal-to-noise characteristics. However, additional studies across a broader range of excipients and process conditions will be necessary to establish the generalizability and robustness of this approach for endpoint detection in industrial settings.

Reviewers comment 1.4 While the manuscript justifies its emphasis on TVIS and Pirani gauges, it could be improved by including a more critical comparison with emerging non-invasive PATs (e.g., wireless temperature sensors, advanced spectroscopic methods

Our response 1.4 : We accept that there are a wide range of  PAT technologies in the literature with new ones being introduced all the time. There is a practical limitation on how many PAT techniques that can be run in a given experiment. We will increase the comparison of  the range of  PAT available and the place of TVIS within this range in the Discussion (see lines 618-640)

Comparison with Emerging Non-Invasive PAT Tools In addition to the techniques evaluated here, a range of emerging non-invasive PAT tools have been proposed for freeze-drying monitoring. Wireless temperature sensors, such as the Tempris® system, offer the advantage of real-time temperature tracking within the product without the cabling challenges of traditional thermocouples, and have shown promise in both pilot and manufacturing scales. However, the temperature profile observed, as the vial warms-up (much like the Pirani profile) presents as a broad curve which asymptotically reaches a plateau when the contents of the vial reaches thermal equilibrium with the shelf. This broad nature of the re-heating warming profile is also an inevitable consequence of the intrinsic overlap between the end of ice sublimation and the developing process of moisture desorption. The point when the profile cross the shelf temperature is often considered the end of ice sublimation, but this is a purely arbitrary stand-point.

Advanced spectroscopic methods, including near-infrared spectroscopy and Raman techniques, have also been explored for monitoring drying dynamics, but their integration into production-scale equipment presents substantial engineering and validation challenges. The actual size dimensions of the sheathed fibre optic probes mean that there will be inevitable consequences for the packing of the vials if these sensors are brought into close proximity to the target vials.

By contrast, TVIS provides a unique combination of non-invasive measurement, sensitivity to the evolving ice mass, and scalability across batches without requiring specialized optical access or proprietary vial designs. The approach therefore occupies a complementary niche within the broader PAT landscape, bridging the gap between temperature-based sensors and indirect chamber pressure measurements.

Reviewers comment 1.5 Specific statistical treatment of results, error analysis, or reproducibility across replicates would strengthen the manuscript.

Our response 1.5 :

The method developed is more qualitative than quantitative in the sense that we have proposed a method for qualifying that

  • an observable inflection in the Pirani profile for larger scale batches is indicative of the ice sublimation end point
  • the mid-point of smaller scale batches is indicative of the ice sublimation end point

Moreover, we would like to stress that it often accepted by many researchers in the freeze-drying sector that the mid-point is always the indicator of the sublimation end point when in fact for a larger scale batch the inflection that we believe to be associated with the sublimation end point can occur before (and maybe in other cases after) the mid-point is passed.

We have added a statement in the Discussion clarifying that this study is exploratory in nature and does not include formal statistical treatment of replicates, and outlining our plans for future quantitative validation (see lines 642-647)

Feasibility and Future Research Directions The present study was designed as a feasibility investigation to explore qualitative correspondences between TVIS-derived endpoints and characteristic features of the Pirani pressure profile. As such, no formal statistical treatment of replicate measurements or detailed error analysis was performed. Future work will incorporate clustered replicates, endpoint reproducibility studies, and quantitative modelling to strengthen the statistical confidence in the proposed interpretations.

Reviewer 2 Report

Comments and Suggestions for Authors

This manuscript presents an innovative and technically solid study proposing the use of through-vial impedance spectroscopy (TVIS) to qualify the Pirani gauge profile as a semi-quantitative tool for determining the end of ice sublimation in freeze-drying. The work is timely and relevant, addressing a well-known challenge in lyophilization process optimization. The methodology is robust, and the results are presented and well-discussed. Some minor revisions are recommended to improve clarity and reinforce validation:

  1. Clarify the definition and operation of TVIS earlier in the manuscript for readers unfamiliar with the technique.

  2. Consider including, or at least discussing, cross-validation of the endpoint using an alternative established technique

Author Response

Reviewers comment 2.1 Clarify the definition and operation of TVIS earlier in the manuscript for readers unfamiliar with the technique.

Our response 2.1 : We refer the referee to lines 117-127 of the original manuscript where the earlier publications using TVIS are referred to and the applications that are opened up by using it.

We have now added an introductory description of the principle and operation of TVIS at the end of the introduction to aid readers unfamiliar with the technique (lines 147-164)

The method involves applying a small alternating voltage across electrodes attached to the outside of the vial and recording the resulting current response. Changes in the impedance spectrum reflect the evolving dielectric properties of the vial contents, which are influenced by the amount of unfrozen water, the structure of the ice matrix, and the progression of drying. In particular, the real part of the measured capacitance at high frequencies above that across which the dielectric relaxation of ice is observed (e.g., in our case 100 kHz) serves as a sensitive indicator of ice mass loss, largely independent of product temperature effects. The distant advantage of this method is that it allows for continuous, real-time assessment of primary drying without introducing probes into the vial headspace or product.

Reviewers comment 2.2 Consider including, or at least discussing, cross-validation of the endpoint using an alternative established technique

Our response 2.2 : The system we used had both pressure rise testing as well  as comparative barometric analysis. The problem with both techniques is that they are sensitive to the water that is evolved form the frozen mass, whether it comes from ice sublimation or moisture desorption.

Similarly, MTM and TDLAS operate based on an assessment of the water vapour that is generated by ice sublimation but also form moisture desorption, and so theoretically these techniques may not help differentiate the end point of ice sublimation from the continued and evolving contribution from water desorption, which results from exposing the internal surfaces as the ice is denuded from the frozen mass.

We have recently investigated, together with another grant partner, a dewpoint humidity analysis, (which has a greater sensitivity to vapour than the Pirani gauge) so in  future we may include this method, to cross-validate the trends we have observed for the Pirani profile. However, this work has not been progressed sufficiently to add that data in this manuscript.  But again, these methods do not differentiate between the two sources of water and so probably wont help much in the validation of the sublimation specific end point.

The only definitive method would be to cluster the TVIS vials at the core of the shelf (to get a more precise determination of the Pirani curve sublimation end point) and to interrupt the cycle just after the TVIS qualified Pirani end point has been passed, and to check for ice melt back. However, that can only be implemented for materials that crystallise and produce a mechanically robust cake that will withstand a temperature ramp to bring the system back to room temperature for the vials to be removed and to observe any wet spots underneath the cake. We are planning to do that in our next programme of work, but this will be dependent on available grant funding.

We have added the following to the Discussion section (lines 603-616)

Cross-Validation with Established PAT Methods Although several established techniques for monitoring freeze-drying progress—such as manometric temperature measurement (MTM), pressure rise testing, and dewpoint humidity analysis—can provide complementary information about the evolution of water vapor in the chamber, all of these methods fundamentally detect total water vapor flux and therefore cannot differentiate between ice sublimation and subsequent moisture desorption from the dried matrix. This limitation reduces their utility for cross-validating the sublimation-specific endpoint identified by TVIS. While preliminary investigations with dewpoint humidity monitoring are underway in our laboratory, the work has not yet progressed to a stage that would allow robust comparison or inclusion here. A more definitive approach would be to interrupt the cycle shortly after the TVIS-determined endpoint and examine the product visually for melt-back or residual ice, but this strategy requires mechanically robust crystalline formulations and dedicated experimental design. We plan to pursue such validation studies in future work.

Reviewer 3 Report

Comments and Suggestions for Authors

Abstract:
1. How was the Pirani-midpoint correlation validated?

  1. It doesn't clearly highlight that this is the first time TVIS has been used to qualify the Pirani profile across batch scales and formulations.
  2. TVIS captures morphological changes in ice during sublimation, and that the 100 kHz signal is sensitive to ice mass but independent of temperature. Explain it.

Introduction:

  1. Provide a brief table or sentence contrasting Pirani vs. MTM vs. TDLAS in terms of Physical basis (thermal conductivity vs. pressure rise), accuracy, scale-up suitability and need for chamber valve control
  2. What is the mechanistic basis for the assumption that the mid-point of the Pirani curve corresponds to the sublimation endpoint?
  3. The manuscript discusses limitations of comparative pressure measurements. Could the authors elaborate on why pressure-rise tests and MTM may fail or be impractical at scale?
  4. Why were edge and core vials chosen as monitoring points?
  5. Can the authors more clearly define what constitutes the “poorly defined transitions” in Pirani-based endpoint detection? Is this based on signal noise, sensor lag, or batch variability?
  6. What specific gap does this study fill?
  7. The principle behind Pirani pressure measurement (differential thermal conductivity sensitivity to water vapor) is briefly mentioned but not technically explained.
  8. The manuscript implies but does not explicitly state what new knowledge this study contributes.

Materials and Methods:

  1. While chemical suppliers are listed, the grade/purity (e.g., USP, Ph. Eur., analytical grade) is missing.
  2. The authors used 100 kHz but don't explain why this frequency is optimal for ice mass sensitivity.
  3. The buffer is described verbally but lacks a formal recipe table or molar concentrations for all excipients.
  4. The software “LyoView®” and “Sciospec analyser” are cited, but versions, calibration procedures, and sampling rate control are not described.
  5. State any EMI shielding or grounding measures used during impedance data acquisition.
  6. Discuss the calibration of TVIS and thermocouples before use.
  7. How were outlier vials handled if one failed or showed abnormal impedance curves?

Results and section of general discussions:

  1. Do you think it is appropriate to include error bars, standard deviations, or replicate variability for TVIS readings or endpoint determinations?
  2. TVIS results are shown as single-trace profiles (Figures 5–8) with no error bars or replicates.
  3. Fit Pirani data to sigmoidal or logistic models.
  4. Although the study aims to correlate TVIS endpoints with the Pirani midpoint, Pirani profiles for these batches are not overlaid or referenced directly in Figure 5.
  5. Figure 6 is not analyzed in parallel with Pirani data for these BSA/MSHT batches.
  6. No Discussion of Formulation-Specific Drying Behavior in Figure 6.
  7. Are t₁ and t₂ chosen manually, algorithmically, or based on signal stability? (Figure 7)
  8. There’s no discussion of how the endpoint shifts with other RMSD thresholds. (Figure 7)
  9. The symmetry or slope of the Pirani curve around its midpoint is visually referenced but not analyzed. (Figure 8)
  10. The actual quantitative time difference between core and edge vial endpoints is not reported. (Figure 8)
  11. Figure 9 lacks TVIS-derived endpoint markers overlaid on the Pirani curves.
  12. The desorption phase is visibly elongated in larger batches, but this is not quantified. (Figure 10)

Author Response

Reviewers comment (general) 3.1 How was the Pirani-midpoint correlation validated?

Our response 3.1 : The mid-point as the end point of ice sublimation was first reported by Pikal in his 2010 paper and this was qualified by a sampling thief removing vials for weighing (to determine loss of ice) and Karl-Fisher to determine residual moisture.

As, we state in response to reviewer 2.2 question the validation of the Pirani end point by an established technique is difficult to achieve. 

Currently, we might claim that TVIS is the only technique that differentiates between moisture desorption and ice sublimation.

Reviewers comment (general) 3.2 It doesn't clearly highlight that this is the first time TVIS has been used to qualify the Pirani profile across batch scales and formulations.

See our response to your comment 3.11 … of which part is reproduced here

This study provides the first systematic demonstration, across multiple formulations and batch sizes, that the end point of ice sublimation determined by TVIS corresponds either to the approximate mid-point or, in larger batches, to an identifiable inflection on the Pirani pressure profile.

Reviewers comment (general) 3.3 TVIS captures morphological changes in ice during sublimation, and that the 100 kHz signal is sensitive to ice mass but independent of temperature. Explain it.

The morphological changes are explained in the paper by Pandya et al (2020) and the independence from temperature is a well-known fact for the dielectric properties of ice as report by Popoov et al.(2017) .

For the temperature independence statement see Lines 322-327

At such frequencies, the dielectric properties of ice are largely temperature independent [24-26] and therefore any change it this parameter can be associated primarily with the mass of ice (rather than its temperature). 

See also our response to your comment 3.13 (in your Methods review section)

Introduction:

Reviewers comment (Introduction) 3.4

Provide a brief table or sentence contrasting Pirani vs. MTM vs. TDLAS in terms of Physical basis (thermal conductivity vs. pressure rise), accuracy, scale-up suitability and need for chamber valve control

We ended up re-structuring the entire introduction in order to respond to various comments form the reviewers and so while we don’t have an explicit section making the requested comparison we think that the introduction now serves its purpose to identify the strengths and weakness of each of these techniques

See new text added in blue at lines 78-83 and 97 - 124

This text also addresses reviewer comment 3.6

Reviewers comment (Introduction) 3.5 What is the mechanistic basis for the assumption that the mid-point of the Pirani curve corresponds to the sublimation endpoint?

In previously published work, there was an observation for sucrose that the mid-point was the end point of sublimation and this was qualified by a sampling thief method that removed selected vials at specific time points during the decent of the Pirani profile/curve. That is what we are reporting on Lines 113-116

Patel et al. [13] compared the shape of the Pirani end of cycle profile with samples removed and measured for residual moisture and showed that the mid-point of the Pirani curve corresponded to the end of ice sublimation, with the remaining part of the curve being defined by moisture desorption form the solids fraction.

Since then, we believe that it is broadly accepted that the mid-point is the sublimation end point. However, our works suggest that this is not always the case.

The only mechanistic proposition is that ice sublimation must come before water desorption, because surfaces of the solids fraction need to be first exposed before desorption can occur. However, its clear form this work that the chamber moisture vs time profile as recorded by the Pirani is a complex function of the water distribution between the ice and the adsorbed fraction and the architecture and porosity of the dry layer that is exposed.

Only computer modelling of ice sublimation and moisture desorption might provide the mechanistic basis for predicting the point on the Pirani curve that corresponds to the end of ice sublimation – but this would require a detailed understanding of the structure of the ice and the interstitial spaces (the solutes fraction) and  is outside the scope of the work and beyond our current capability.  We could explore the modelling work in future via our collaboration with Siemens.

In the introduction we already have a reference to some work that was done in this regard (see line 145-146)

Fontana et al. [18] described simulation tools for predicting primary drying endpoint in industrial development.

To this end we have added the following to lines 559-570 of the revised manuscript

Mechanistic Basis of Pirani Mid-Point Interpretation  It is important to note that the common practice of interpreting the approximate mid-point of the Pirani profile as corresponding to the end of ice sublimation is primarily based on earlier empirical studies, particularly the work of Patel et al. [13], who demonstrated this relationship in sucrose formulations using direct sampling and moisture assays. Mechanistically, this convention is rationalized by the principle that ice sublimation must necessarily precede the onset of significant water desorption, as desorption requires exposure of the dried matrix surface. The present results demonstrate that the Pirani gauge profile is a complex convolution of sublimation dynamics, desorption kinetics, and evolving product resistance. A definitive mechanistic model capable of predicting the precise transition point would require simulation approaches, coupled with detailed data on product microstructure and porosity, and is beyond the scope of this study.

Reviewers comment (Introduction) 3.6 The manuscript discusses limitations of comparative pressure measurements. Could the authors elaborate on why pressure-rise tests and MTM may fail or be impractical at scale?

See response to 3.4

Reviewers comment (Introduction) 3.7 Why were edge and core vials chosen as monitoring points?

As referred to in referee 1 response, the largest likely variation in the drying across a shelf especially in a dryer with a Perspex chamber door  is the difference in sublimation rate due to the impact of radiant heat. Hence by covering both points we have acquired data on these trends. 

Reviewers comment (Introduction) 3.8 Can the authors more clearly define what constitutes the “poorly defined transitions” in Pirani-based endpoint detection? Is this based on signal noise, sensor lag, or batch variability?

The “poorly defined transitions” are not primarily due to signal noise, sensor lag, or random batch variability. Instead, they arise because the shape of the Pirani pressure profile inherently lacks clear inflection points or discontinuities at the true end of ice sublimation.

The reason why the Pirani profile has no sharp transition, is because as ice sublimation slows down, water desorption from the dried matrix gradually becomes the dominant source of water vapour in the chamber. This process is continuous, not stepwise. As a consequence, the curve shows a smooth, sigmoid-like decay in apparent pressure with no obvious “kink” that can be unambiguously attributed to the moment ice is fully gone. In effect, the transition “poorly defined” because there is no discrete signature (e.g., sudden drop or inflection) in the signal that uniquely marks the end of sublimation. Instead, the signal drifts continuously into the desorption phase.

While noise, lag, or variability can compound the uncertainty, the fundamental problem is that the signal itself is a superposition of two overlapping phenomena (sublimation + desorption), neither of which ends abruptly.

In summary, the poorly defined transition is primarily due to the absence of clear features (like inflections) in the shape of the Pirani pressure profile, rather than measurement noise, sensor lag, or batch-to-batch variability.

We state ‘poorly-defined’ in the abstract so don’t have an opportunity (because of word count) to add further explanation there. So instead we have included this text in the manuscript to explain this  issue succinctly. See line 117-124 of the revised manuscript

Consequently, the Pirani pressure profile typically assumes a broadly sigmoid shape: an initial phase dominated by ice sublimation, followed by a phase increasingly influenced by water desorption. The transition between these phases—marking the completion of sublimation—is often approximated near the midpoint of the curve. However, due to the smooth and continuous nature of the Pirani signal, lacking sharp inflections or discontinuities, this transition is often ill-defined in practical settings, complicating the use of Pirani data for precise endpoint detection.

Reviewers comment (Introduction) 3.9 What specific gap does this study fill?

The specific gap is that nobody has systematically used TVIS to map and define where, on the smooth Pirani curve, the true end of sublimation occurs—so that the Pirani gauge can be repurposed as a semi-quantitative production-scale tool.

Lots of people rely on Pirani profiles, but they guess (or conservatively overextend drying times) because there is no validated reference point. Our study correlate TVIS (which can directly detect the ice disappearance) with the Pirani profile to define this reference point. This makes it possible to trust the Pirani plateau/midpoint as a reliable marker—something that has not been robustly demonstrated across batches and formulations.

We have added the following text to line 182-187 the revised manuscript

To date, no systematic approach has been established to directly correlate non-invasive, vial-scale measurements of ice sublimation—such as those provided by through-vial impedance spectroscopy—with the characteristic but continuous Pirani pressure profiles recorded at chamber scale. This gap has limited the ability to use the Pirani gauge as a robust, semi-quantitative marker of sublimation endpoint in production environments, resulting in reliance on conservative cycle extensions or costly alternative sensors.

Reviewers comment (Introduction) 3.10 The principle behind Pirani pressure measurement (differential thermal conductivity sensitivity to water vapor) is briefly mentioned but not technically explained.

Some additional text has been added on lines 101-107 of the revised manuscript

The operating principles of a Pirani gauge is that it estimates the chamber pressure based on the thermal conductivity of the surrounding gas. As the chamber pressure increases, the heated filament within the gauge loses heat more efficiently to the gas, resulting in a change in electrical resistance that is calibrated to indicate pressure. Because water vapour has significantly higher thermal conductivity than dry gases such as nitrogen, even small amounts of water vapour lead to an overestimation of the true chamber pressure.

Reviewers comment (Introduction) 3.11 The manuscript implies but does not explicitly state what new knowledge this study contributes.

Results: “The take home message is that the sublimation end-point of the TVIS vials appear suggest that it is the mid-point of the Pirani curve that represents the batch end of the sublimation process. Thereafter, the remaining position of the Pirani curve must de facto be a consequence of the water desorption from the solids fraction.”

This is the clearest statement of what is new, i.e., we are providing evidence correlating the TVIS sublimation end point with the mid-point of the Pirani curve, across different formulations and batch sizes.

General Discussion: “Time points from TVIS determined sublimation end points suggest that it is the mid-point of the Pirani curve that corresponds to the batch end point of sublimation. This is in agreement with [7] in their study on 5% sucrose.”

Here were are in agreement with Patel et al., so we are partially confirming prior findings.

Conclusion: “Results to date show promise that TVIS may qualify the point on the Pirani curve that signifies the end of ice sublimation across the batch…”

Again we are proposing that TVIS can validate the location on the Pirani curve that corresponds to sublimation completion, which is the central contribution.

Where the text does explicitly claim new knowledge:

“…as the batch size increases, an inflection in the profile (coupled to the observed TVIS determined sublimation end points) suggests that the profile may then be divided into two distinct regions…”

Here we are suggesting new evidence that in large batches, the Pirani profile becomes more structured (shows an inflection), allowing easier demarcation between sublimation and desorption. And so the end of ice sublimation isn’t necessarily defined by the mid-point of the tram-lines between the high value of the Pirani measurement and the low value of the Pirani measurement

To strengthen the manuscript in this regard we have added the following to the conclusion (lines 652-670)

This study provides the first systematic demonstration, across multiple formulations and batch sizes, that the end point of ice sublimation determined by TVIS corresponds either to the approximate mid-point or, in larger batches, to an identifiable inflection on the Pirani pressure profile. Importantly, the observations indicate that this inflection does not necessarily align with the mid-point. For the larger batches, the second part of the Pirani profile is characterised by a long, drawn-out tail resulting from moisture desorption. If the inflection occurs before the mid-point (as in the BSA formulation studied here), reliance on the mid-point necessarily delays the decision point for switching to the secondary drying ramp and results in longer-than-necessary cycles. Notably, the extended second part of the Pirani curve is evident even in materials such as mannitol that crystallise and thus typically carry less adsorbed water, although in this case partial hydrate formation may contribute to the effect. This behaviour is not necessarily apparent for smaller batches of mannitol, which show a precipitous decrease after the mid-point. Our study therefore suggests a basis for interpreting Pirani signals as semi-quantitative markers of sublimation completion without recourse to invasive probes, thereby addressing a critical gap in freeze-drying process monitoring at scale. Extending these observations to larger commercial dryers could allow optimization of production processes and reduce reliance on conservative waiting periods to de-risk the switch to secondary drying.

Materials and Methods:

Reviewers comment (Materials and Methods) 3.12

While chemical suppliers are listed, the grade/purity (e.g., USP, Ph. Eur., analytical grade) is missing.

The purity complies with PhEur for sucrose,  mannitol, and L-histidine (sourced from Merck)

Tween 20 was sourced from BioRad ( low peroxide grade   product 1706531)

BSA was Fraction V powder,  cell culture grade from PAN-Biotech UK  (Wimborne, UK)

Sterile water for irrigation was  from Baxter product UKF7114  ( Baxter Healthcare, Zurich, Switzerland).

Reviewers comment (Materials and Methods) 3.13

The authors used 100 kHz but don't explain why this frequency is optimal for ice mass sensitivity.  

The reason is that the dielectric permittivity of ice, at these relatively high frequencies, is largely temperature independent, and therefore the impact of the inevitable warming of the vial, as sublimation completes makes no or little contribution to the shape of the profile, and in particular the sharpness of the end point. If temperature did have an effect beyond the end of ice sublimation within an individual vial then it would result in the capacitance continuing to increase rather than attaining a  plateau.

To address this we add the following after this part of the manuscript

“In this case, a frequency of 100 kHz was selected, with the value of the real part capacitance thereafter defined by the term C′(100 kHz). At such frequencies, the dielectric properties of ice are largely temperature independent [24–26] and therefore any change in this parameter can be associated primarily with the mass of ice (rather than its temperature).” Lines 300-310

This characteristic is essential for reliably determining the sublimation end point, because as sublimation progresses, the product temperature inevitably increases. At lower frequencies, the dielectric properties of ice are more strongly dependent on temperature, which can lead to an artificial rise in the measured capacitance as the frozen matrix warms. Such an effect might obscure the transition to the plateau phase and reduce the precision of end point detection. By contrast, at 100 kHz, any temperature-induced changes are negligible, ensuring that the recovery of the capacitance reflects the shrinking of the ice dome across the base of the vial, rather than a temperature artifact. This frequency, or indeed any frequency between 100 kHz and 1 MHz, which is the upper limit of the frequency range of the current analyser, is therefore optimal for isolating the influence of ice mass on the signal profile.

Reviewers comment (Materials and Methods) 3.14

The buffer is described verbally but lacks a formal recipe table or molar concentrations for all excipients.

Its not clear why a formal table would be required when the text description provides all necessary detail.

Reviewers comment (Materials and Methods) 3.15

The software “LyoView®” and “Sciospec analyser” are cited, but versions, calibration procedures, and sampling rate control are not described.

Software and Sciospec S/N added . Detail of calibrations added to lines 302-316. Sampling rate already provided on line 294 (see also response to your comment 3.17)

Reviewers comment (Materials and Methods) 3.16

State any EMI shielding or grounding measures used during impedance data acquisition.

We carried out all the impedance measurements within the stainless steel chamber of the freeze dryer and earthed the impedance analyser to the mains water copper piping in the laboratory

We have added the following to the methods (lines 264-267)

All impedance measurements were performed within the stainless-steel chamber of the freeze-dryer, which effectively acts as a Faraday cage to minimize electromagnetic interference (EMI). To further reduce electrical noise, the impedance analyser GND port was connected to the laboratory’s mains water piping (copper).

Reviewers comment (Materials and Methods) 3.17

Discuss the calibration of TVIS and thermocouples before use.

The TVIS channels were individually calibrated using a 100fF reference standard on the day of the cycle was started

Checks on the quality of spectra form each vial were undertaken with low conductivity water  in each vial before the TVIS readings on the run were made.

The thermocouples were not calibrated routinely per se, but have a check point taken at ambient temperature.

Answered along with 3.18

Reviewers comment (Materials and Methods) 3.18

How were outlier vials handled if one failed or showed abnormal impedance curves?

If any vial failed the calibration or failed to show an impedance peak as expected with water then all connections were checked  and the vial re-calibrated. Only vials which showed an acceptable impedance water spectrum were used in the run.

We have added the following to the methods, after we explain how you attached the electrodes and calibrated the system.

We have added the following to the methods (lines 268-278)

Each TVIS channel was calibrated immediately prior to use with a combination of an open circuit element and then a 100 fF reference standard, both supplied by LyosenZ Ltd as part of their TVIS installation package). In addition, the impedance spectrum of each connected vial was verified using water for irrigation to confirm the presence of a characteristic dielectric relaxation peak before starting the experimental cycle. Any vial that failed to exhibit a characteristic impedance spectrum for pure water (i.e., showing a distinct dielectric relaxation peak in the imaginary part capacitance spectrum) was removed from the array, reconnected, and re-calibrated. Only vials that demonstrated an acceptable impedance profile during this pre-run validation were included in the experiment. If calibration or connection issues persisted after re-calibration, the vial was excluded from the run to avoid introducing spurious data.

Thermocouples were not routinely calibrated on a per-run basis but were periodically verified by recording their readings at ambient laboratory temperature and comparing them against a calibrated reference thermometer to confirm consistent baseline accuracy.

Results and General Discussion:

Reviewers comment (Results and Discussion) 3.19

Do you think it is appropriate to include error bars, standard deviations, or replicate variability for TVIS readings or endpoint determinations?

In future, we could look to a more comprehensive study whereby we cluster the 5 TVIS vials at specific locations, e.g., 5 at the front on one run, and 5 at the core on another separate run, and then average and take standard deviations of the end points of each step of 5 at the specific locations.

To this end we add the following to the Methods where we describe the positioning the vials (lines 227-235)

In this feasibility study, each TVIS channel was connected to a single vial positioned at a defined location (e.g., front, core, edge) within the batch. Consequently, the variability between vials at the same location was not assessed, and no error bars are presented. While this design allows a qualitative comparison of drying behaviour across positions, a more robust statistical characterization of measurement reproducibility will require multiple TVIS vials clustered at identical locations within the same batch to enable calculation of standard deviations and confidence intervals for estimated endpoints. Future work will adopt this approach to quantify intra-location variability and assess the precision of the method.

Reviewers comment (Results and Discussion) 3.20

TVIS results are shown as single-trace profiles (Figures 5–8) with no error bars or replicates.

See 3.19

We refer the reviewer to the new text in the Methods section (paragraph X), where we explicitly state that each TVIS channel was connected to a single vial and that no intra-location variability was assessed. This design was chosen to demonstrate feasibility rather than provide statistical reproducibility, which is planned for future work.

Reviewers comment (Results and Discussion) 3.21

Fit Pirani data to sigmoidal or logistic models. 

See response to 3.27

Reviewers comment (Results and Discussion) 3.22

Although the study aims to correlate TVIS endpoints with the Pirani midpoint, Pirani profiles for these batches are not overlaid or referenced directly in Figure 5.

Reviewers comment (Results and Discussion) 3.23

Figure 6 is not analyzed in parallel with Pirani data for these BSA/MSHT batches.

We shall address comment 3.22 and 3.23 together

Each figure in the manuscript is introduced step by step, in order to walk the reader through the various observations that can be made, as we lead up to divulging the method, which culminates in the overlaying of the TVIS end points on the Pirani curve. Which is ultimately what we present in Figures 9 and 10.

Figure 5 and 6 for that matter highlight another interesting observation along the way, which is the differences and similarities of the time it takes for the dome of the ice mass (for the centre vial in all the batches) to shrink from covering the entire base to the point at which it has shrunk to nothing.

Reviewers comment (Results and Discussion) 3.34

No Discussion of Formulation-Specific Drying Behavior in Figure 6.

Thank you for the prompt to explain/comment on the observation that BSA vials show similar recovery times across batch sizes, while mannitol vials show slower recovery rates in larger batches. Its not clear if this is significant and so we add the following at lines 362-365.

With the limited data set under scrutiny it is not yet clear whether these differences in the rate of recovery for the mannitol batches is significant and whether it points towards some structural differences in the frozen mass. .

Reviewers comment (Results and Discussion) 3.25

Are t and t chosen manually, algorithmically, or based on signal stability? (Figure 7)

In this study, t₁ and t₂ were selected manually, informed by visual inspection of the C′(100 kHz) profiles. Specifically, we identified a segment within the late-stage drying data where the signal appeared to have reached a stable plateau, as indicated by a consistent slope near zero and low variability in the raw capacitance measurements. This approach was adopted pragmatically, given the heterogeneity in drying behavior across formulations and batch sizes and the exploratory nature of the work. We acknowledge that algorithmic or automated criteria for defining t₁ and t₂ (e.g., based on statistical measures of signal stability) would enhance reproducibility and objectivity, and we plan to develop and validate such methods in future studies. For transparency, the selected intervals for each experiment are documented in the supplementary materials.

We have also added the following to the revised manuscript on lines 383-389

For each drying run, the time points t₁ and t₂ defining the regression interval were selected manually based on visual inspection of the C′(100 kHz) profiles. The aim was to identify a segment within the later part of primary drying where the signal had reached a stable plateau, as indicated by a consistently low slope and minimal variability in the capacitance measurements. This pragmatic approach was adopted due to the substantial variation in drying behaviour observed across different formulations and batch sizes, which precluded applying a uniform algorithmic criterion. While this manual selection introduces an element of subjectivity, it enabled consistent identification of the plateau region across experiments. Future work will focus on developing automated or statistical methods to define these intervals objectively and improve reproducibility.

Reviewers comment (Results and Discussion) 3.26

There’s no discussion of how the endpoint shifts with other RMSD thresholds. (Figure 7)

The optimization of the specific number of RMSDs requires extensive additional studies to build a statistical picture of how TVIS maps onto the Pirani end point, which is currently outside the scope of what we can achieve currently.

For now we can make some general observations, that by increasing the number of RMSD’s to three, the end point is shifted to earlier time points, with the front vial end point occupying a position on the plateau prior to the step down. We therefore made the assumption that 3 RMSD’s was too many. Similarly, reducing the number of RMSDs to 1 shifted the end point to beyond the mid-point for centre vials, which does no support the accepted concept that the mid-point is the sublimation end point for smaller batches and so then assumed that 1 RMSD was too few.

To this end we have added the following text to lines 400-415 of the revised manuscript

We conducted exploratory analyses to evaluate the sensitivity of the estimated end point to the RMSD criterion. Specifically, increasing the threshold from 2.2 RMSD to 3 RMSD systematically shifted the identified end points earlier in time, often to positions on the plateau of the C′(100 kHz) profile that preceded the characteristic “step down” transition observed in the front vials. Conversely, reducing the threshold to 1 RMSD consistently delayed the estimated end points beyond the mid-point of the Pirani profile in core vials. This outcome conflicts with prior findings (e.g., Patel et al. [7]) indicating that the approximate mid-point of the Pirani curve represents the end of ice sublimation for smaller batch sizes. Based on these comparative observations, we selected 2.2 RMSD as a practical compromise that balances sensitivity and specificity in the context of this study.

The optimization of the RMSD threshold used to retrospectively identify the sublimation end point warrants a dedicated study with a larger sample size and rigorous statistical modelling, And so future work will systematically assess alternative RMSD thresholds to build a more robust statistical mapping between TVIS-determined end points and Pirani profiles across diverse formulations and scale conditions

Reviewers comment (Results and Discussion) 3.27

The symmetry or slope of the Pirani curve around its midpoint is visually referenced but not analyzed. (Figure 8)

In this study, the description of the slope and symmetry of the Pirani curve around its midpoint was based on qualitative visual inspection of the time-pressure profiles across different batch sizes and formulations. We agree that a quantitative analysis—such as calculating the first derivative of the Pirani signal to assess the slope, or fitting parametric sigmoid or biexponential models to formally characterize symmetry—would provide a more rigorous assessment of these observations. However, the primary objective of this work was to demonstrate the feasibility of correlating TVIS-derived sublimation endpoints with identifiable positions on the Pirani curve, rather than to comprehensively model curve shape parameters. Future studies with larger datasets and dedicated design will systematically analyse the slope and symmetry around the identified transition point to evaluate their potential as predictive markers.

To his end we have added the following text to lines 431-436of the revised manuscript

While the slope and symmetry of the Pirani curves were noted qualitatively, formal quantitative modelling of these features was beyond the scope of this initial study and will be addressed in future work. In addition, the quantitative assessment of the duration and slope of the desorption phase was beyond the scope of this study but will be pursued in future work to better characterize its dependence on batch size and formulation properties.

Reviewers comment (Results and Discussion) 3.28

The actual quantitative time difference between core and edge vial endpoints is not reported

We have added these to a table ??

Reviewers comment (Results and Discussion) 3.29

Figure 9 lacks TVIS-derived endpoint markers overlaid on the Pirani curves.

TVIS endpoints as a single point  were presented  for greater clarity. The principle of connecting the TVS end point by a line to the Pirani curve was demonstrated in Fig. 8

Reviewers comment (Results and Discussion) 3.30

The desorption phase is visibly elongated in larger batches, but this is not quantified. (Figure 10)

In this study, we qualitatively noted that the desorption phase of the Pirani profile appeared more gradual and prolonged in larger batches, reflecting the extended release of bound water as the batch size increased. While these trends were consistently observed, we acknowledge that we did not formally quantify the duration or slope of the desorption phase across batches, nor did we define explicit phase boundaries that would enable systematic measurement. The primary focus of this work was to establish the correspondence between TVIS-determined sublimation endpoints and features of the Pirani curve, rather than to develop a comprehensive quantitative characterization of the desorption kinetics. However, we agree that formal analysis of the desorption phase (e.g., calculation of its time span and rate of pressure decline) would strengthen the interpretation of these results, and we intend to address this in future studies by applying consistent operational definitions and regression models to the desorption segment.

See response 3.27.